# Extensive intraspecies cryptic variation in an ancient embryonic gene regulatory network

Yamila N Torres Cleuren[1,4,2†], Chee Kiang Ewe[1,4], Kyle C Chipman[1,4], Emily R Mears[2], Cricket G Wood[1,4], Coco Emma Alma Al-Alami[2], Melissa R Alcorn[1,4], Thomas L Turner[3], Pradeep M Joshi[1,4], Russell G Snell[2], Joel H Rothman[1,2,3,4]*

[1]Department of MCD Biology, University of California, Santa Barbara, Santa Barbara, United States; [2]School of Biological Sciences, University of Auckland, Auckland, New Zealand; [3]Department of Ecology, Evolution, and Marine Biology, University of California, Santa Barbara, Santa Barbara, United States; [4]Neuroscience Research Institute, University of California, Santa Barbara, Santa Barbara, United States

**Abstract** Innovations in metazoan development arise from evolutionary modification of gene regulatory networks (GRNs). We report widespread cryptic variation in the requirement for two key regulatory inputs, SKN-1/Nrf2 and MOM-2/Wnt, into the *C. elegans* endoderm GRN. While some natural isolates show a nearly absolute requirement for these two regulators, in others, most embryos differentiate endoderm in their absence. GWAS and analysis of recombinant inbred lines reveal multiple genetic regions underlying this broad phenotypic variation. We observe a reciprocal trend, in which genomic variants, or knockdown of endoderm regulatory genes, that result in a high SKN-1 requirement often show low MOM-2/Wnt requirement and *vice-versa*, suggesting that cryptic variation in the endoderm GRN may be tuned by opposing requirements for these two key regulatory inputs. These findings reveal that while the downstream components in the endoderm GRN are common across metazoan phylogeny, initiating regulatory inputs are remarkably plastic even within a single species.
DOI: https://doi.org/10.7554/eLife.48220.001

*For correspondence:
joel.rothman@lifesci.ucsb.edu

Present address: †Department of Informatics, Computational Biology Unit, University of Bergen, Bergen, Norway

Competing interests: The authors declare that no competing interests exist.

## Introduction

While the core regulatory machinery that specifies embryonic germ layers and major organ identity in the ancestor of modern animals is largely conserved in all extant animals, GRN architecture must be able to accommodate substantial plasticity to allow for evolutionary innovation in developmental strategies, changes in selective pressures, and genetic drift (*Peter and Davidson, 2011*; *Félix and Wagner, 2008*). Genetic variation, often with neutral effects on fitness, provides for plasticity in GRN structure and implementation (*Félix and Wagner, 2008*). Although studies of laboratory strains of model organisms with a defined genetic background have been highly informative in identifying the key regulatory nodes in GRNs that specify developmental processes (*Davidson and Levine, 2008*; *Oliveri et al., 2008*; *Peter and Davidson, 2017*), these approaches do not reveal the evolutionary basis for plasticity in these networks. The variation and incipient changes in GRN function and architecture can be discovered by analyzing phenotypic differences resulting from natural genetic variation present in distinct isolates of a single species (*Milloz et al., 2008*; *Nunes et al., 2013*; *Phinchongsakuldit et al., 2004*).

**eLife digest** Two people with the same disease, or who inherit the same genetic mutation, often show different symptoms or respond to medical treatments in different ways. This is because many traits are not the result of a single gene, but of several genes interacting with each other in complex ways to form networks that lead to many possible outcomes.

Gene regulatory networks, which control how animals develop, change over evolutionary time to create the vast variety of different species that exist today. However, it is still unclear how mutations in these networks can occur without negatively impacting their activity, or how networks become rewired during evolution. To address these questions, Torres Cleuren et al. studied the gene regulatory network that controls the development of the gut across approximately 100 different strains of *Caenorhabditis elegans*, a widely studied nematode worm. This involved testing how switching off particular genes affected gut development in embryos of the worm.

The experiments revealed that the first steps in the gene regulatory networks that control gut development vary drastically between the different wild strains of *C. elegans*. For example, in some of the strains, two genes known as *skn-1* and *mom-2* are essential for gut formation, whereas in others the gut often forms even when these genes are switched off. These results support the idea that some of the genes in the network can compensate for loss of others, explaining how mutations can accumulate without impacting the development of the embryo.

The findings of Torres Cleuren et al. provide important insights into how gene regulatory networks can be rewired, with some components accumulating mutations and acquiring new roles, while others stay the same.

DOI: https://doi.org/10.7554/eLife.48220.002

The endoderm has been proposed to be the most ancient of the three embryonic germ layers in metazoans (*Hashimshony et al., 2015*; *Rodaway and Patient, 2001*), having appeared prior to the advent of the Bilateria about 600 Mya (*Peterson et al., 2004*). It follows, therefore, that the GRN for endoderm in extant animals has undergone substantial modifications over the long evolutionary time span since its emergence. However, the core transcriptional machinery for endoderm specification and differentiation appears to share common mechanisms across metazoan phylogeny. For example, cascades of GATA-type transcription factors function to promote endoderm development not only in triploblastic animals but also in the most basal metazoans that contain a digestive tract (*Martindale et al., 2004*; *Boyle and Seaver, 2008*; *Boyle and Seaver, 2010*; *Gillis et al., 2007*; *Davidson et al., 2002*). Among the many observations supporting a common regulatory mechanism for establishing the endoderm, it has been found that the endoderm-determining GATA factor, END-1, in the nematode *C. elegans*, is sufficient to activate endoderm development in cells that would otherwise become ectoderm in Xenopus (*Shoichet et al., 2000*). This indicates that the role of GATA factors in endoderm development has been preserved since the nematodes and vertebrates diverged from a common ancestor that lived perhaps 600 Mya.

To assess the genetic basis for evolutionary plasticity and cryptic variation underlying early embryonic germ layer specification, we have analyzed the well-described GRN for endoderm specification in *C. elegans*. The E cell, which is produced in the very early *C. elegans* embryo, is the progenitor of the entire endoderm, which subsequently gives rise exclusively to the intestine. The EMS blastomere at the four-cell stage divides to produce the E founder cell and its anterior sister, the MS founder cell, which is the progenitor for much of the mesoderm (*Sulston et al., 1983*). Both E and MS fates are determined by maternally provided SKN-1, an orthologue of the vertebrate Nrf2 bZIP transcription factor (*Bowerman et al., 1992*; *Bowerman et al., 1993*; *Maduro and Rothman, 2002*). In the laboratory N2 strain, elimination of maternal SKN-1 function (through either knockdown or knockout) results in fully penetrant embryonic lethality as a result of misspecification of EMS cell descendants. In these embryos, the fate of MS is transformed to that of its cousin, the mesectodermal progenitor C cell. E cells similarly adopt a C cell-like fate in a majority, but not all, of these embryos (*Bowerman et al., 1992*). SKN-1 initiates mesendoderm development via the GRN in E and MS cells in part by activating zygotic expression of the MED-1/2 divergent GATA transcription factors (*Maduro et al., 2007*; *Maduro et al., 2001*). This event mobilizes a cascade of GATA factors in the

E cell lineage that ultimately direct intestinal differentiation (*Maduro and Rothman, 2002*; *Maduro, 2017*; *Wiesenfahrt et al., 2016*; *McGhee, 2007*).

This differential requirement for SKN-1 in endoderm (E) and mesoderm (MS) development is determined by its combinatorial action with triply redundant Wnt, MAPK, and Src signaling systems, which act together to polarize EMS (*Meneghini et al., 1999*; *Shin et al., 1999*; *Bei et al., 2002*; *Thorpe et al., 1997*). MOM-2/Wnt acts through the MOM-5/Frizzled receptor, mobilizing WRM-1/β-catenin, resulting in its cytoplasmic accumulation in the posterior side of EMS. WRM-1, together with LIT-1/NLK kinase, alters both the nucleocytoplasmic distribution and activity of the Wnt effector POP-1/Tcf (*Thorpe et al., 1997*; *Nakamura et al., 2005*; *Rocheleau et al., 1999*), converting it from a repressor of endoderm in the MS cell lineage to an activator in the E cell lineage (*Owraghi et al., 2010*; *Huang et al., 2007*; *Maduro et al., 2002*; *Phillips et al., 2007*; *Maduro et al., 2005a*; *Shetty et al., 2005*). Loss of MOM-2 expression in the laboratory N2 strain results in a partial gutless phenotype, while removal of both MOM-2 and SKN-1, through either knockdown or knockout, leads to a completely penetrant loss of gut (*Thorpe et al., 1997*), revealing their genetically redundant roles.

The regulatory relationship between SKN-1 and POP-1, the effector of Wnt signaling, shows substantial variation even in species that diverged 20–40 million years ago, suggesting significant evolutionary plasticity in this key node in the endoderm GRN. *C. elegans* embryos lacking maternal POP-1 always make gut, both in the normal E cell lineage and in the MS cell lineage. However, in embryos lacking both SKN-1 and POP-1, endoderm is virtually never made, implying that these two factors constitute a Boolean 'OR' logic gate. In contrast, removal of either SKN-1 or POP-1 alone in *C. briggsae* causes >90% of embryos to lack gut, indicative of an 'AND' logic gate (*Figure 1A,B*) (*Lin et al., 2009*).

In this study, we sought to determine whether the plasticity in regulatory logic of the two major inputs into endoderm development is evident within a single species. The availability of many naturally inbred variants (isotypes) of *C. elegans* that show widespread genomic variation (*Félix and Braendle, 2010*; *Cook et al., 2017*; *Andersen et al., 2012*), provides a genetically rich resource for investigating potential quantitative variation in developmental GRNs. We report here that the requirement for activation of the endoderm GRN by SKN-1 or MOM-2, but not POP-1, is highly variable between natural *C. elegans* isolates, and even between closely related isotypes. Genome-wide association studies (GWAS) in isolates from the natural populations and targeted analysis of recombinant inbred lines (RILs), revealed that a multiplicity of loci and their interactions are responsible for the variation in the developmental requirement for SKN-1 and MOM-2. We found a complex, but frequently reciprocal requirement for SKN-1 and MOM-2 among variants underlying these phenotypes: loci associated with a high requirement for SKN-1 in endoderm development tend to show a more relaxed requirement for MOM-2 and *vice-versa*. Consistent with this finding, three other endoderm regulatory factors, RICT-1, PLP-1, and MIG-5, show similar inverse relationships between these two GRN inputs. These findings reveal that the activation of the GRN network for a germ layer, one of the most critical early developmental switches in embryos, is subject to remarkable genetic plasticity within a species and that the dynamic and rapid change in network architecture reflects influences distributed across many genetic components that affect both SKN-1 and Wnt pathways.

## Results

### Extensive natural cryptic variation in the requirement for SKN-1 in endoderm specification within the *C. elegans* species

The relationship between SKN-1 and Wnt signaling through POP-1 in the endoderm GRN has undergone substantial divergence in the *Caenorhabditis* genus (*Lin et al., 2009*). While neither input alone is absolutely required for endoderm specification in *C. elegans*, each is essential in *C. briggsae*, which has been estimated to have diverged from *C. elegans* ~ 20–40 Mya (*Zhao et al., 2008*; *Cutter, 2008*). In contrast to the *C. elegans* N2 laboratory strain, removal of either SKN-1 or POP-1 alone results in fully penetrant conversion of the E founder cell fate into that of the mesectodermal C blastomere and of E to MS fate, respectively, in *C. briggsae* (*Lin et al., 2009*). These findings revealed that the earliest inputs into the endoderm GRN are subject to substantial evolutionary differences between these two species (*Figure 1B*). We sought to determine whether incipient

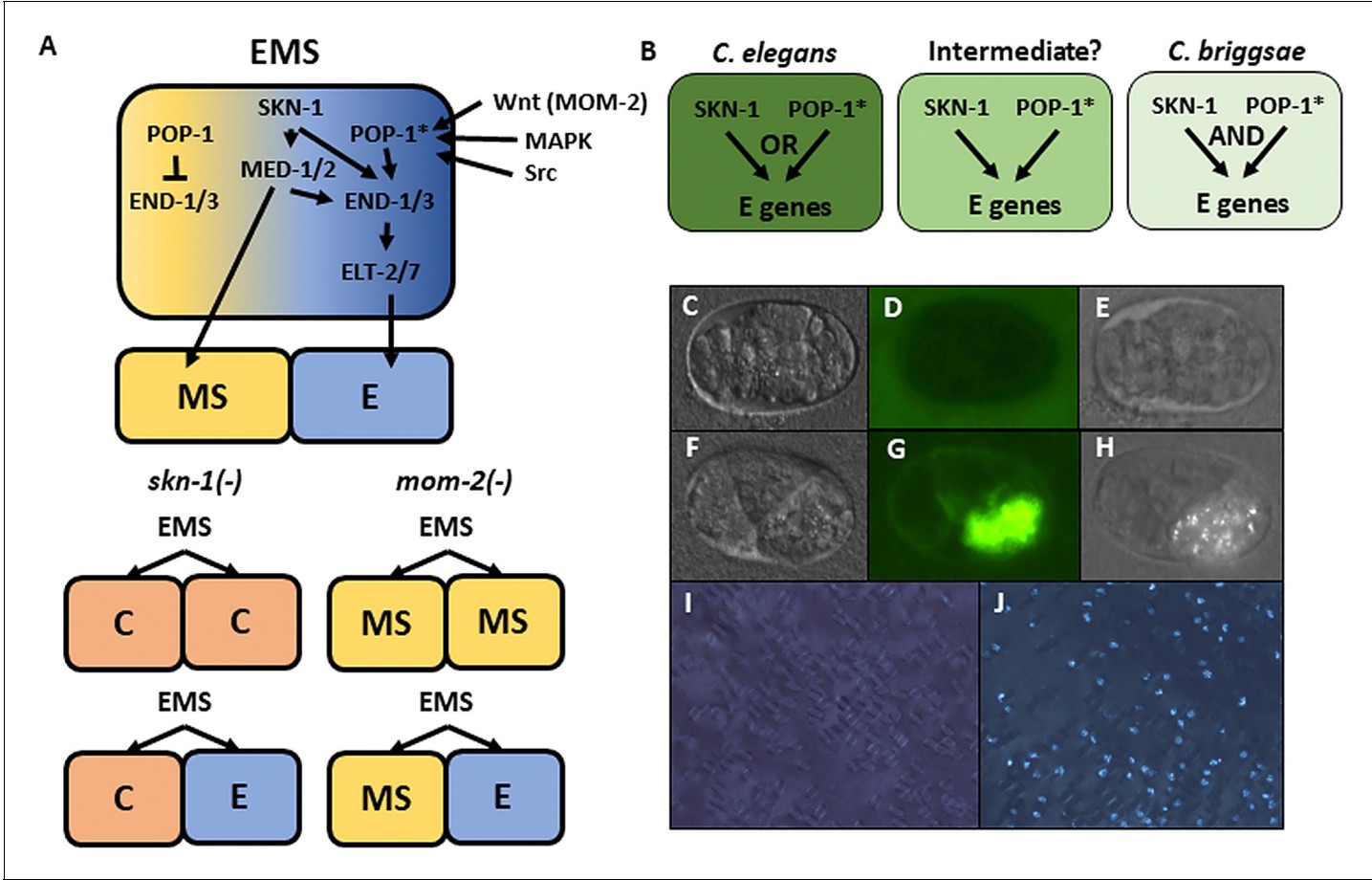

**Figure 1.** Endoderm regulatory pathway and scoring of gut differentiation. (**A**) Under normal conditions, signaling from the posterior $P_2$ cell (Wnt, MAPK and Src) results in asymmetric cortical localization of Wnt signaling pathway components in EMS leading to POP-1 asymmetry in the descendants of EMS, with high levels of nuclear POP-1 in anterior MS and low levels of nuclear POP-1 in the posterior, E, daughter cell. In the anterior MS cell, high nuclear POP-1 represses the END genes, allowing SKN-1 to activate MS fate. In the posterior E cell, which remains in contact with P2, POP-1 is converted to an activator and, along with SKN-1, activates the END genes, resulting in endoderm fate. Loss of *skn-1,* either by RNAi or in loss-of-function mutants, causes 100% of the embryos to arrest; in 70% of the arrested embryos, EMS gives rise to two C-like cells, while in the remaining 30% only MS is converted to a C fate; the posterior daughter retains its E fate. Loss of *mom-2* leads to 100% embryonic arrest with a partially penetrant E→MS cell fate transformation, resulting in two MS-like daughter cells in ~72% of the embryos. (**B**) Regulatory logic of SKN-1 and POP-1 in E specification in *C. elegans, C. briggsae* and a hypothetical intermediate state. POP-1* denotes the activated state. (**C-H**) Gut visualization in embryos affected by *skn-1* RNAi. (**C-E**) arrested embryos without endoderm, (**F-H**) arrested embryos with endoderm. (**C, F**) DIC images of arrested embryos ~ 12 hr after egg laying. (**D, G**) the same embryos expressing the gut-specific *elt-2::GFP* reporter, and (**E,H**) birefringent gut granules under polarized light. All embryos showing gut birefringence also show *elt-2::GFP* expression. (**I, J**) Fields of arrested *skn-1(RNAi)* embryos in wild isolate strains JU1491 (**I**) and JU440 (**J**), which reflect the extremes in the spectrum of requirement of SKN-1 in gut development at 0.9% and 60%, respectively.

DOI: https://doi.org/10.7554/eLife.48220.003

evolutionary plasticity in this critical node at the earliest stages of endoderm development might be evident even within a single species of the *Caenorhabditis* genus by assessing their requirement in *C. elegans* wild isolates and testing whether the quantitative requirements of each input were correlated.

Elimination of detectable maternal SKN-1 from the laboratory N2 strain by either a strong (early nonsense) chromosomal mutation (*skn-1(zu67)*), or by RNAi knockdown, results in a partially penetrant phenotype: while the E cell adopts the fate of the C cell in the majority of embryos, and gut is not made,~30% of arrested embryos undergo strong gut differentiation, as evidenced by the appearance of birefringent, gut-specific rhabditin granules, or expression of *elt-2::GFP*, a marker of the developing and differentiated intestine (*Figure 1C–H*). In our experimental conditions, we found that RNAi of *skn-1* in different N2-derived mutant strains gave highly reproducible results: 100% of

the embryos derived from *skn-1(RNAi)*-treated mothers arrest (n > 100,000) and 32.0 ± 1.9% of the arrested embryos exhibited birefringent gut granules (*Figure 2A*; *Supplementary file 1*) over many trials by separate investigators. We found that the LSJ1 laboratory strain, which is derived from the same original source as N2, but experienced very different selective pressures in the laboratory owing to its constant propagation in liquid culture over 40 years (*Sterken et al., 2015*), gave virtually identical results to that of N2 (31.0% ± s.d 1.2%), implying that SKN-1-independent endoderm formation is a quantitatively stable trait. The low variability in this assay, and high number of embryos that can be readily examined (≥500 embryos per experiment), provides a sensitive and highly reliable system with which to analyze genetic variation in the endoderm GRN between independent *C. elegans* isolates.

To assess variation in SKN-1 requirement within the *C. elegans* species, we analyzed the outcome of knocking down SKN-1 by RNAi in 96 unique *C. elegans* wild isolates (*Andersen et al., 2012*). Owing to their propagation by self-fertilization, each of the isolates (isotypes) is a naturally inbred clonal population that is virtually homozygous and defines a unique haplotype. The reported estimated population nucleotide diversity averages $8.3 \times 10^{-4}$ per bp (*Andersen et al., 2012*), and we found that a substantial fraction (29/97) of isotypes were quantitatively indistinguishable in phenotype between the N2 and LSJ1 laboratory strains (*Figure 2A*, *Supplementary file 1*). We found that all strains, with the exception of the RNAi-resistant Hawaiian CB4856 strain, invariably gave 100% embryonic lethality *with skn-1(RNAi)*. However, we observed dramatic variation in the fraction of embryos with differentiated gut across the complete set of strains, ranging from 0.9% to 60% (*Figure 2A*). Repeated measurements with >500 embryos per replicate per strain revealed very high reproducibility (*Figure 2—figure supplement 1*), indicating that even small differences in the fraction of embryos generating endoderm could be reproducibly measured. Further, we found that some wild isolates that were subsequently found to have identical genome sequences also gave identical results.

Although birefringent and autofluorescent rhabditin granules have been used as a marker of gut specification and differentiation in many studies (*Clokey and Jacobson, 1986*; *Hermann et al., 2005*), it is conceivable that the variation in fraction of embryos containing this marker that we observed might reflect variations in gut granule formation rather than in gut differentiation per se. We note that embryos from all strains showed a decisive 'all-or-none' phenotype: that is, they were either strongly positive for gut differentiation or completely lacked gut granules, with virtually no intermediate or ambiguous phenotypes. A threshold of gene activity in the GRN has been shown to account for such an all-or-none switch in gut specification (*Maduro et al., 2007*; *Raj et al., 2010*; *Zhu et al., 1997*). This observation is inconsistent with possible variation in gut granule production: if SKN-1-depleted embryos were defective in formation of the many granules present in each gut cell, one might expect to observe gradations in numbers or signal intensity of these granules between gut cells or across a set of embryos. Nonetheless, we extended our findings by analyzing the expression of the gut-specific intermediate filament IFB-2, a marker of late gut differentiation, in selected strains representing the spectrum of phenotypes observed (*Figure 2B*). As with gut granules, we found that embryos showed all-or-none expression of IFB-2. In all cases, we found that the fraction of embryos containing immunoreactive IFB-2 was not significantly different (Fisher's exact test, p-values>0.05) from the fraction containing gut granules, strongly suggesting that the strains vary in endoderm specification per se and consistent with earlier studies of SKN-1 function (*Bowerman et al., 1992*; *Maduro et al., 2007*).

Although we found that *skn-1(RNAi)* was 100% effective at inducing embryonic lethality in all strains (with the exception of the RNAi-defective Hawaiian strain, CB4856), it is conceivable that, at least for the strains that showed a weaker phenotype than for N2 (i.e., higher number of embryos specifying endoderm), the variation observed between strains was attributable to differences in RNAi efficacy rather than in the endoderm GRN. Indeed, studies with N2 and CB4856 showed that germline RNAi sensitivity is a quantitative trait, involving the Argonaute-encoding *ppw-1* gene and additional interacting loci present in some wild isolates (*Tijsterman et al., 2002*; *Pollard and Rockman, 2013*). To address this possibility, we introgressed the strong loss-of-function *skn-1(zu67)* chromosomal mutation into five wild isolates whose phenotypes spanned the spectrum observed (ranging from 2% of embryos with differentiated gut for MY16% to 50% for MY1) (*Figure 2C*). In all cases, we found that introgression of the allele through five rounds of backcrosses resulted in a quantitative phenotype that was similar or indistinguishable from that observed with *skn-1(RNAi)*.

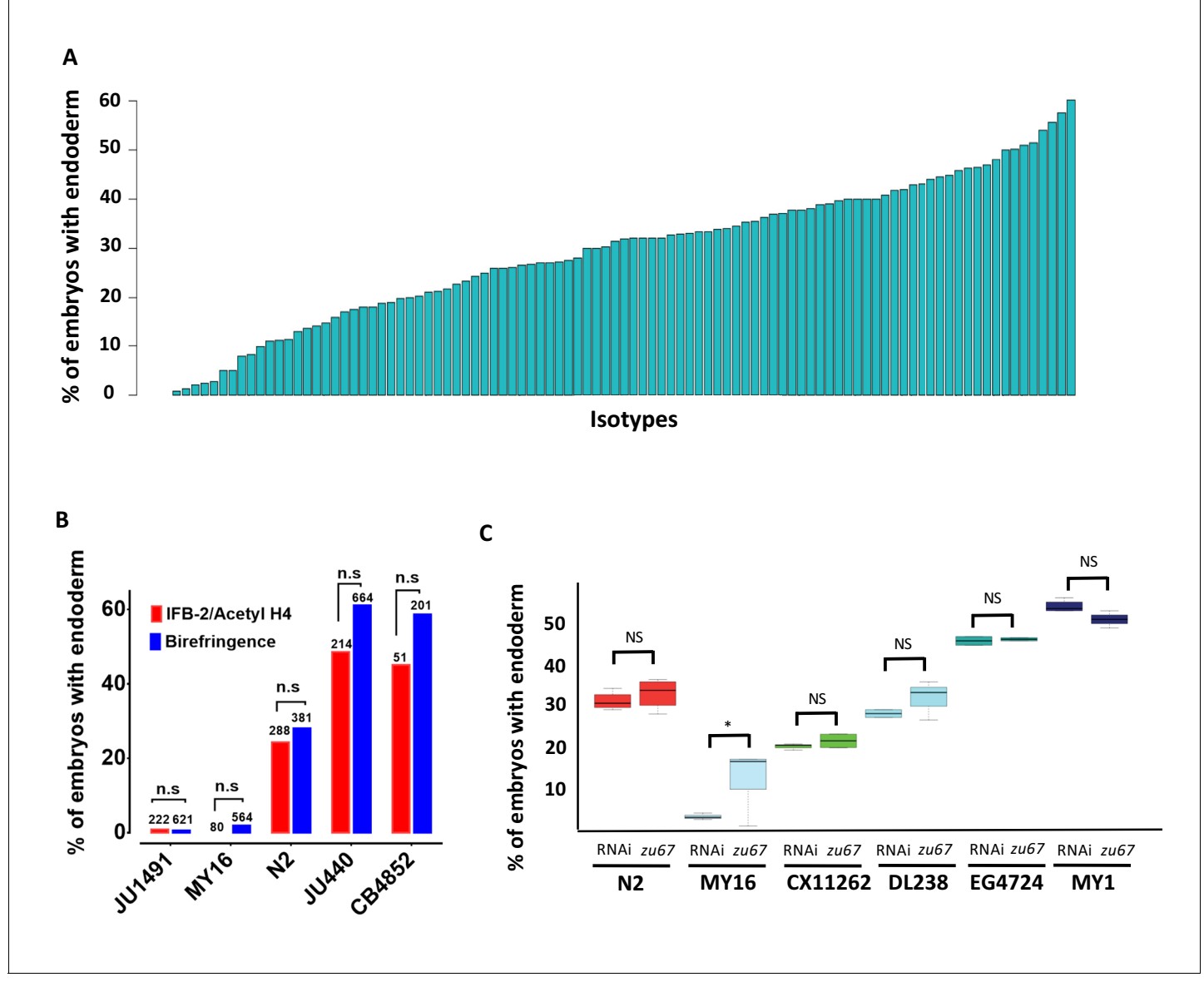

**Figure 2.** Quantitative effects of loss of *skn-1* on endoderm formation. (**A**) Spectrum of *skn-1(RNAi)* effects across the *C. elegans* isolates. The effects of *skn-1(RNAi)* are quantified as the average percentage of arrested embryos with endoderm (y-axis). All wild isolates treated with *skn-1(RNAi)* resulted in 100% embryonic arrest (n > 500 embryos per replicate per isotype and at least two replicates per isotype). (**B**) Comparison of *skn-1(RNAi)* phenotype using two different gut markers (birefringent gut granules and MH33 staining of IFB-2) in five different genetic backgrounds. In all cases, no significant statistical difference was found between the two quantitative methods. Fisher's exact test (NS p-value>0.05). (**C**) Comparison of *skn-1(RNAi)* and *skn-1 (zu67)* effects on endoderm development in six different genetic backgrounds. For each color-coded strain, the first value is of the *skn-1(RNAi)* results (five replicates), while the second is the result for the *skn-1(zu67)* allele introgression (10 replicates). For all strains (with the exception of MY16), no significant statistical difference was found between the RNAi knockdown and corresponding *skn-1(zu67)* allele effects on endoderm development. Student t-test (NS p-value>0.05, * p-value<0.05).

DOI: https://doi.org/10.7554/eLife.48220.004

The following figure supplement is available for figure 2:

**Figure supplement 1.** High reproducibility *of skn-1(RNAi)* phenotypes in various *C. elegans* isotypes.

DOI: https://doi.org/10.7554/eLife.48220.005

The phenotypes of the introgressed allele were significantly different (p-values<0.01) from that of the parental N2 *skn-1(zu67)* strain, except for DL238, whose *skn-1(RNAi)* phenotype was indistinguishable from that of N2. The results obtained by introgression from four of the isotypes (CX11262, DL238, EG4724 and MY1), were not statistically different (Student t-test, p-values>0.05) from the

corresponding RNAi knockdown results (*Figure 2C*) (i.e., the phenotype was suppressed or enhanced relative to N2 in these genetic backgrounds to the same extent as with *skn-1(RNAi))*. However, while the MY16 *skn-1(zu67)* strain shifted in the predicted direction (i.e., became stronger) as compared to the N2 strain, it showed a weaker phenotype than was evident by RNAi knockdown, even following eight rounds of introgression. Regardless, diminished RNAi efficacy in MY16 cannot explain the large difference between the *skn-1(RNAi)* phenotype of N2 and MY16, as the latter phenotype is, in fact, much stronger, not weaker, than the former. As described below, we identified a modifier locus in the MY16 strain that is closely linked to the *skn-1* gene; it therefore seems likely that the N2 chromosomal segment containing this modifier was carried with the *skn-1(zu67)* mutation through the introgression crosses, thereby explaining the somewhat weaker phenotype of the introgressed allele in MY16. We conclude that the extreme variation in *skn-1(RNAi)* phenotype between the wild isolates tested results from *bona fide* cryptic variation in the endoderm GRN, rather than differences in RNAi efficacy.

We note that the strength of *skn-1(RNAi)* phenotype does not correlate with phylogenetic relatedness between the strains (Pagel's λ = 0.42, p-value=0.14). For example, while some closely related strains (e.g., MY16 and MY23) showed a similar phenotype, other very closely related strains (e.g., JU1491 and JU778) showed phenotypes on the opposite ends of the phenotypic spectrum (*Figure 3A*). We also did not observe any clear association between geographical distribution and *skn-1 (RNAi)* phenotype (*Figure 3B*). These findings suggest that the initiating inputs into the endoderm GRN is subject to rapid intraspecies evolutionary divergence.

## Cryptic variation in the quantitative requirement for MOM-2/Wnt, but not POP-1, in endoderm development

The switch in the relationship of the SKN-1 and Wnt inputs between *C. elegans* ('OR' operator) and *C. briggsae* ('AND' operator) (*Lin et al., 2009*), and the extensive variation in the requirement for SKN-1 seen across *C. elegans* isolates, raised the possibility that the quantitative requirement for Wnt components might vary between unique isolates of *C. elegans*. It has been shown that signaling from Ras pathway varies in different *C. elegans* wild isolates and hyperactive Wnt signaling can compensate for reduced Ras activity in the vulva signaling network (*Milloz et al., 2008*; *Gleason et al., 2002*). Given that removal of the maternal Wnt input also results in a partially penetrant gut defect (through either knock-out or knockdown of Wnt signaling components), it is conceivable that a compensatory relationship may exist between the SKN-1 and Wnt inputs. We investigated this possibility by examining the requirement for the MOM-2/Wnt ligand in the same wild isolates that were tested for the SKN-1 gut developmental requirement. Indeed, we observed broad variation in the requirement for MOM-2/Wnt in activation of the endoderm GRN between isotypes. *mom-2(RNAi)* of 94 isotypes resulted in embryonic arrest, indicating that, as with *skn-1(RNAi)*, *mom-2(RNAi)* was effective at least by the criterion of lethality. Two isotypes, CB4853 and EG4349, did not exhibit *mom-2(RNAi)*-induced lethality and were omitted from further analyses. In the affected strains, the fraction of *mom-2(RNAi)* embryos with differentiated gut varied from ~40% to~99% (*Figure 4A*, *Supplementary file 1*). As with *skn-1(RNAi)*, the *mom-2(RNAi)* phenotype of isotypes N2, JU440, and JU1213 was further confirmed by immunostaining with IFB-2 (*Figure 4B*), again demonstrating that birefringence of gut granules is a reliable proxy for endoderm formation for this analysis.

To assess whether the observed variation in the *mom-2(RNAi)* phenotype reflected differences in the GRN or RNAi efficacy, the *mom-2(or42)* allele was introgressed into three different genetic backgrounds chosen from the extreme ends of the phenotypic spectrum. *mom-2(RNAi)* of the laboratory N2 strain resulted in the developmental arrest of embryos. Of those,~90% contained differentiated endoderm, a result that was highly reproducible. In contrast, the introgression of an apparent loss-of-function allele, *mom-2(or42)*, into the N2 strain results in a more extreme phenotype: only ~28% of embryos show endoderm differentiation (*Figure 4C*) (*Thorpe et al., 1997*). While this discrepancy can partly be explained by incomplete RNAi efficacy, it is notable that the penetrance of *mom-2* alleles vary widely (*Thorpe et al., 1997*). We observed strain-specific variation in embryonic lethality response to RNAi of *mom-2* between the different isotypes. However, we found that the *mom-2 (or42)* introgressed strains show qualitatively similar effects to those observed with *mom-2* RNAi. For example, the *mom-2(or42)* allele introgressed into the isotype JU1213 background resulted in a low fraction of arrested embryos with gut (5.7% ± s.d 2.4%; n = 2292), a more extreme effect than was seen with RNAi (34.0% ± s.d 1.5%; n = 1876). This is the strongest phenotype that has been reported

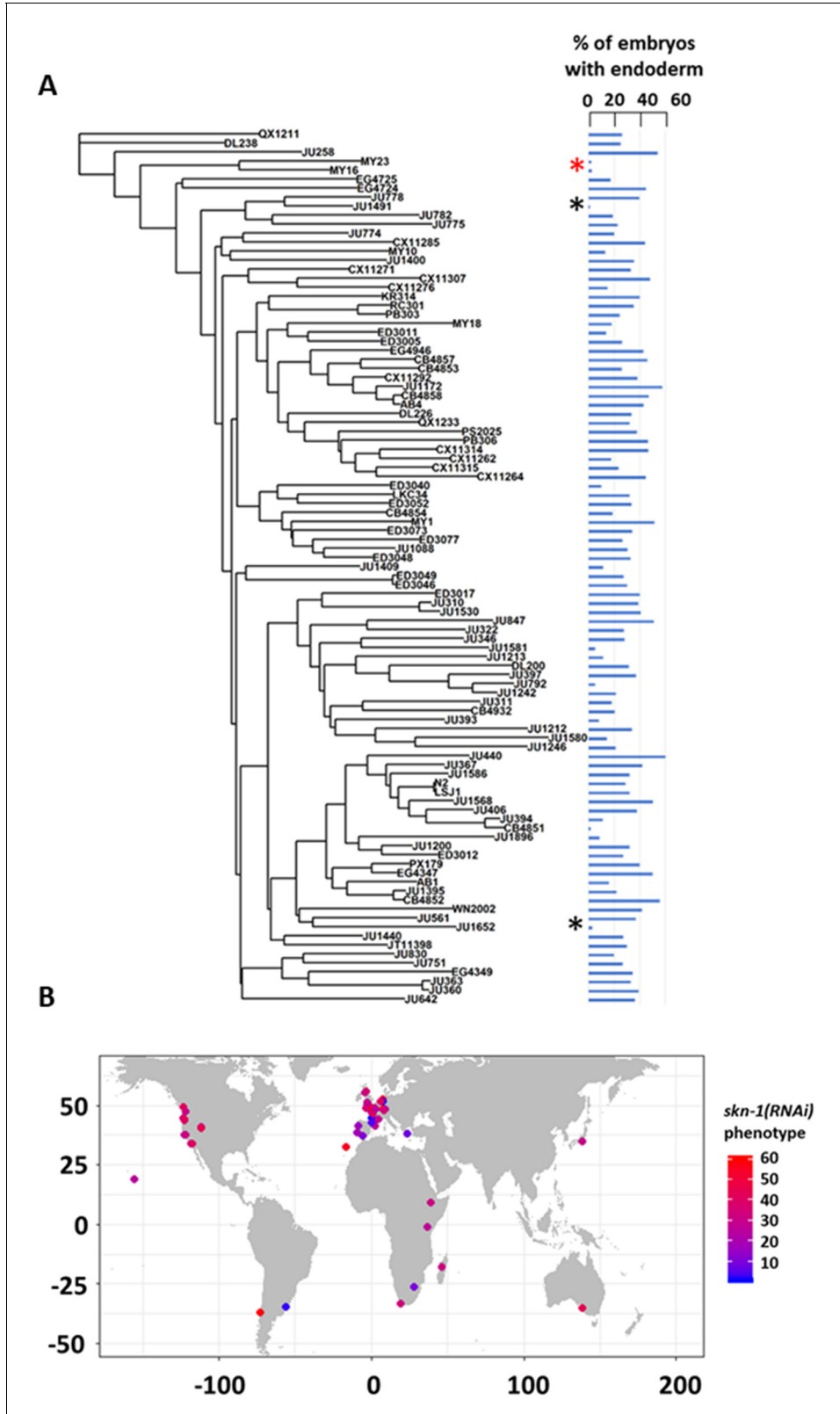

**Figure 3.** SKN-1 requirement does not correlate with genotypic relatedness or geographical location. (A) *skn-1(RNAi)* phenotype of 97 isolates arranged with respect to the neighbor-joining tree constructed using 4,690 SNPs and pseudo-rooted to QX1211. Red asterisk indicates an example of closely related strains (MY23 and MY16) with similar phenotype, while black asterisks indicate example sister strains (JU778 and JU1491; JU561 and

*Figure 3 continued on next page*

*Figure 3 continued*

JU1652) with distinct phenotype. Phylogenetic relatedness and phenotype are not significantly correlated (Pagel's λ = 0.42, p-value=0.14). (B) Worldwide distribution of *skn-1(RNAi)* phenotype across 97 wild isolates. Each circle represents a single isotype.
DOI: https://doi.org/10.7554/eLife.48220.006

for any known *mom-2* allele. On the other hand, introgression of the *mom-2* mutation gave rise to a significantly higher fraction of embryos with endoderm in isotypes DL226 (55.2% ± s.d 1.2%, n = 1377) and PB303 (65.5% ± s.d 4.9%, n = 2726), relative to the laboratory strain N2 (29.1% ± s.d 3.1%; n = 1693), consistent with the RNAi phenotypes (*Figure 4C*). These findings indicate that the differential requirement for MOM-2 is, at least in part, attributable to genetic modifiers in these strains. As with *skn-1(RNAi),* we found no correlation between the *mom-2(RNAi)* phenotype and phylogenetic relatedness or geographical distribution (*Figure 4—figure supplement 1*), suggesting rapid intraspecies developmental system drift.

As the MOM-2/Wnt signal is mediated through the POP-1 transcription factor, we sought to determine whether the requirement for POP-1 might also vary between isolates. We found that, while *pop-1(RNAi)* resulted in 100% embryonic lethality across all 96 RNAi-sensitive isolates, 100% of the arrested embryos contained a differentiated gut (n > 500 for each isolate scored) (results not shown). Thus, all isolates behave similarly to the N2 strain with respect to the requirement for POP-1. These results were confirmed by introgressing a strong loss-of-function *pop-1(zu189)* allele into four wild isolates (N2, MY16, JU440, and KR314) (*Figure 4—figure supplement 2*). The lack of variation in endoderm specification after loss of POP-1 is not entirely unexpected. As has been observed in a *pop-1(-)* mutant strain, elimination of the endoderm-repressive role of POP-1 in the MS lineage (which is not influenced by the P2 signal) supersedes its endoderm activating role in the presence of SKN-1. Indeed, the original observation that all *pop-1(-)* embryos in an N2 background contain gut masked the activating function for POP-1, which is apparent only in the absence of SKN-1 (*Owraghi et al., 2010*; *Maduro et al., 2002*; *Maduro et al., 2005a*). It is likely that, as with the N2 strain, gut arises from both E and MS cells in all of these strains; however, as we have scored only for presence or absence of gut, it is conceivable that the E lineage is not properly specified in some strains, a possibility that cannot be ruled out without higher resolution analysis.

Our results contrast with those of *Paaby et al. (2015)*, who reported that RNAi of 29 maternal-effect genes across a set of 55 wild isolates in liquid culture resulted in generally weaker effects on lethality than we observed. This difference is likely attributable to diminished and variable RNAi efficacy in the latter study owing to the different culture methods used (see Materials and methods) (*Çelen et al., 2018*; *Gomez-Amaro et al., 2015*). To assess this possibility further, we compared our results with those of *Paaby et al. (2015)* and found no correlation between the variation in fraction of embryos with gut and the lethality observed in that report with both *mom-2(RNAi)* and *skn-1 (RNAi)* (Pearson's R = 0.19, p=0.23; Pearson's R = 0.22, p=0.17, respectively). In addition, *Paaby et al. (2015)* found that the genetically divergent strain QX1211 consistently showed weak penetrance across all targeted genes, while under our experimental conditions, QX1211 exhibited a slightly stronger *skn-1(RNAi)* phenotype (25.2% *vs.* 32.0%, Fisher's exact test p-value=0.03) and a similar *mom-2(RNAi)* phenotype (90% *vs.* 90%, Fisher's exact test p-value=0.9) compared to the N2 strains with fully penetrant lethality in all cases.

## Genome-wide association studies (GWAS) and analysis of RILs identify multiple genomic regions underlying variation in the two major endoderm GRN inputs

We sought to examine the genetic basis for the wide variation in SKN-1 and Wnt requirements across *C. elegans* isolates and to evaluate possible relationships in the variation seen with the SKN-1 and Wnt inputs by performing GWAS using the available SNP markers and map (*Andersen et al., 2012*), adjusting for population structure by using Efficient Mixed-Model Analysis (EMMA) (*Figure 5A,B*) (*Kang et al., 2008*; *Wang, 2002*). This approach identified two significant closely-located positions on chromosome IV that underlie the variation in SKN-1 requirement (*Figure 5A*, *Table 1*).

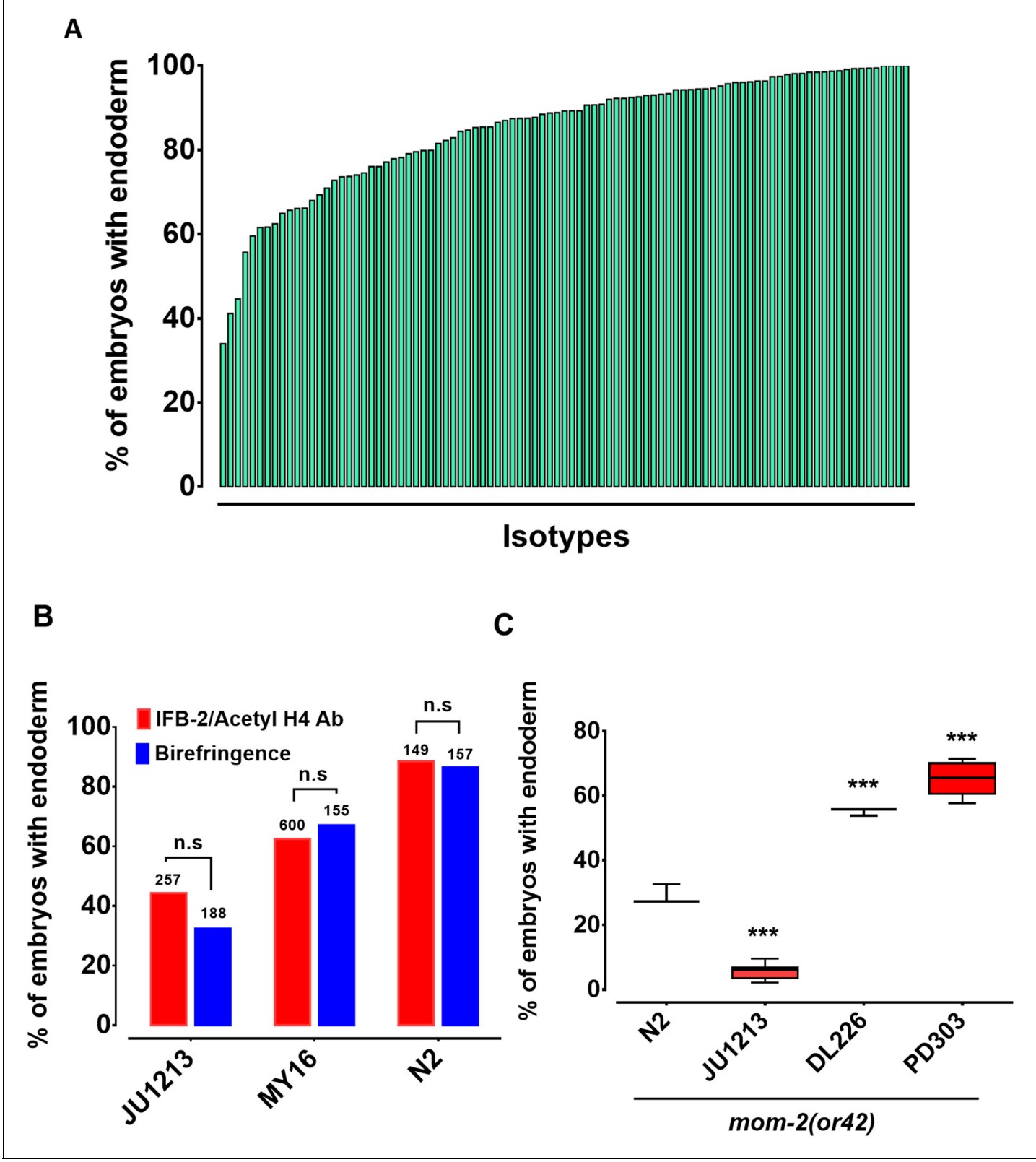

**Figure 4.** Widespread variation in the *mom-2(RNAi)* phenotype. (A) Spectrum of *mom-2(RNAi)* effects across the *C. elegans* isolates. The effects of *mom-2(RNAi)* are quantified as the average percentage of arrested embryos with endoderm (y-axis). Each column represents the mean for each wild isolate (n > 500 embryos were scored for each experiment with at least two replicates per isotype). (B) Comparison of *mom-2(RNAi)* phenotype using two different gut markers (birefringent gut granules and MH33 immunostaining of IFB-2) in three different genetic backgrounds. In all cases, no

*Figure 4 continued on next page*

*Figure 4 continued*

significant statistical difference was found between the two quantitative methods. Fisher's exact test (NS p-value>0.05). (**C**) Comparison of the effect of *mom-2(or42)* on endoderm development after introgression into four different genetic backgrounds. At least three independent introgressed lines were studied for each wild isotype. The results were compared *to N2; mom-2(or42)*. Student t-test (*** p-value<0.001).
DOI: https://doi.org/10.7554/eLife.48220.007
The following figure supplements are available for figure 4:

**Figure supplement 1.** MOM-2 requirement does not correlate with genotypic relatedness or geographical location.
DOI: https://doi.org/10.7554/eLife.48220.008
**Figure supplement 2.** The requirement for POP-1 in endoderm formation does not vary in three introgressed strains.
DOI: https://doi.org/10.7554/eLife.48220.009

GWAS of the *mom-2(RNAi)* variation proved more challenging because this phenotype showed a highly skewed distribution (Shapiro-Wilk' test W = 0.8682, p-value=1.207×10$^{-7}$) (*Figure 5—figure supplement 1*). While GWAS did not reveal any genomic regions for the *mom-2(RNAi)* variation that

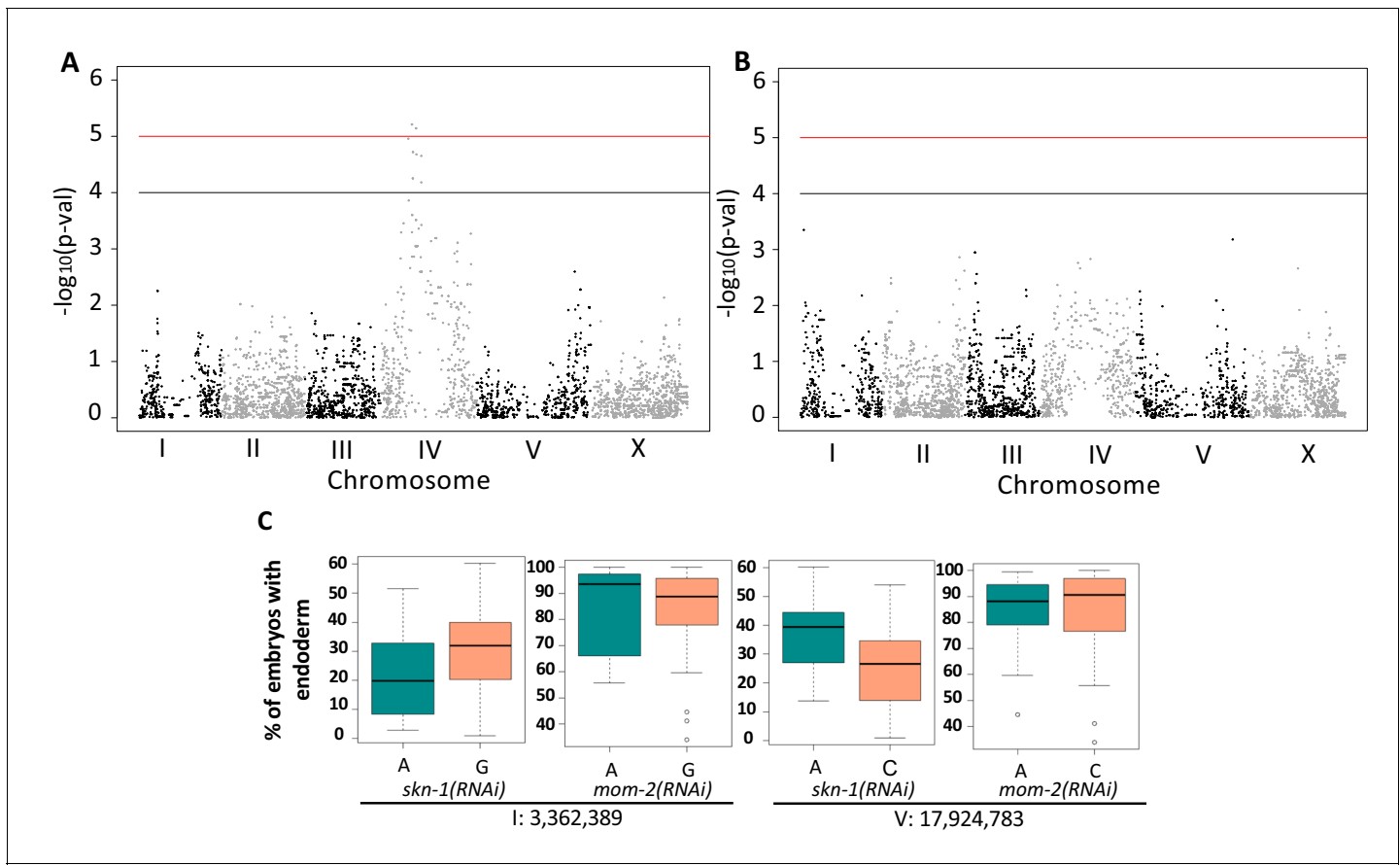

**Figure 5.** Genome-Wide Association Studies of *skn-1(RNAi)* and *mom-2(RNAi)* phenotypes. (**A**) Manhattan plot of *skn-1(RNAi)* GWAS. The red line indicates a genome-wide 1.5% FDR (permutation-based FDR, from 10,000 permutated results). Black line represents 3.0% FDR. The y axis is the –log$_{10}$ of p-value. (**B**) Manhattan plot of *mom-2 (RNAi)* EMMA. The y axis is the –log10 of p-value. Genomic regions are shown on the x-axis. (**C**) Effect plots of the most strongly-linked SNPs from *mom-2(RNAi)* GWAS at position 3,362,389 bp on chromosome I and position 17,924,783 bp on chromosome V. Horizontal lines within each box represent the median, and the boxes represent 25th–75th percentile.
DOI: https://doi.org/10.7554/eLife.48220.010
The following figure supplements are available for figure 5:

**Figure supplement 1.** Histogram of *mom-2(RNAi)* phenotype among the 94 wild isolates.
DOI: https://doi.org/10.7554/eLife.48220.011
**Figure supplement 2.** Comparison of EMMA p-values for both *mom-2* and *skn-1 RNAi* phenotypes.
DOI: https://doi.org/10.7554/eLife.48220.012

**Table 1.** Significantly linked SNPs for *skn-1(RNAi)* GWAS.

| SNP | EMMA -log(p) | N2 allele | Variant allele |
| --- | --- | --- | --- |
| IV: 5,079,371 bp | 4.957651645 | A | G |
| IV: 5,725,367 bp | 5.211140897 | C | T |
| IV: 5,761,153 bp | 5.211140897 | G | A |
| IV: 5,891,378 bp | 4.252324884 | G | A |
| IV: 5,920,597 bp | 4.720037892 | T | A |
| IV: 5,921,302 bp | 4.720037892 | T | G |
| IV: 5,921,510 bp | 4.252324884 | C | T |
| IV: 6,453,892 bp | 5.142174312 | T | A |
| IV: 6,511,989 bp | 5.142174312 | C | A |
| IV: 6,563,740 bp | 4.678423021 | C | T |
| IV: 7,453,945 bp | 4.652004517 | G | A |
| IV: 7,453,143 bp | 4.181579989 | A | G |

DOI: https://doi.org/10.7554/eLife.48220.013

exceeded an FDR of 5%, we found that the most strongly associated loci for the *mom-2(RNAi)* phenotype also showed large effects for *skn-1(RNAi)* (*Figure 5C*). In particular, we observed substantial overlap in the p-values for individual SNPs from *skn-1(RNAi)* and *mom-2(RNAi)* in the central region of chromosome IV (*Figure 5—figure supplement 2*), raising the possibility that common genetic factors might underlie these phenotypes.

In an effort to narrow in on causal loci underlying the *skn-1(-)* and *mom-2(-)* phenotypic variation, and to assess possible relationships between these two GRN inputs, we prepared and analyzed 95 recombinant inbred lines (RILs) between two *C. elegans* isotypes, N2 and MY16. These strains were chosen for their widely varying differences in requirement for both inputs (see Materials and methods). In contrast to the very low variation seen between multiple trials of each parental strain, analysis of the RNAi-treated RIL strains (>500 embryos/RIL) revealed a very broad distribution of phenotypes. We found that, while some RILs showed phenotypes similar to that of the two parents, many showed intermediate phenotypes and some were reproducibly more extreme than either parent, indicative of transgressive segregation (*Rieseberg et al., 2003*). For *skn-1(RNAi)*, the phenotype varied widely across the RILs, with 1% to 47% of embryos containing gut (*Figure 6A*, *Supplementary file 2*). This effect was even stronger with *mom-2(RNAi)*, for which virtually the entire possible phenotypic spectrum was observed across a selection of 31 RILs representing the span of *skn-1(RNAi)* phenotypes. The *mom-2(RNAi)* phenotypes ranged from RILs showing 3% of embryos with gut to those showing 92% (*Figure 6A*). In all RILs, *skn-1(RNAi)* and *mom-2(RNAi)* resulted in 100% lethality. It is noteworthy that one RIL (JR3572, *Supplementary file 2*) showed a nearly completely penetrant gutless phenotype, an effect that is much stronger than has been previously observed for *mom-2(-)* (*Thorpe et al., 1997*). These results indicate that a combination of natural variants can nearly eliminate a requirement for MOM-2 altogether, while others make it virtually essential for endoderm development. Collectively, these analyses reveal that multiple quantitative trait loci (QTL) underlie SKN-1- and MOM-2-dependent endoderm specification.

To identify QTLs from the recombinant population, we performed linkage mapping for both phenotypes using interval mapping (see Materials and methods). For *skn-1(RNAi)*, two major peaks were revealed on chromosomes II and IV (above 1% FDR estimated from 1000 permutations). Two minor loci were found on chromosomes I and X (suggestive linkage, above 20% FDR) (*Figure 6B*). For *mom-2(RNAi)*, two major independent QTL peaks were found on chromosomes I and II (above the 5% FDR estimated from 1000 permutations). Although the candidate peaks observed on chromosome IV for *skn-1(RNAi)* did not appear to overlap with those for *mom-2(RNAi)*, overlap was observed between the chromosomes I and II candidate regions for these two phenotypes (*Figure 6B*). These QTLs show large individual effects on both phenotypes (*Figure 6C*).

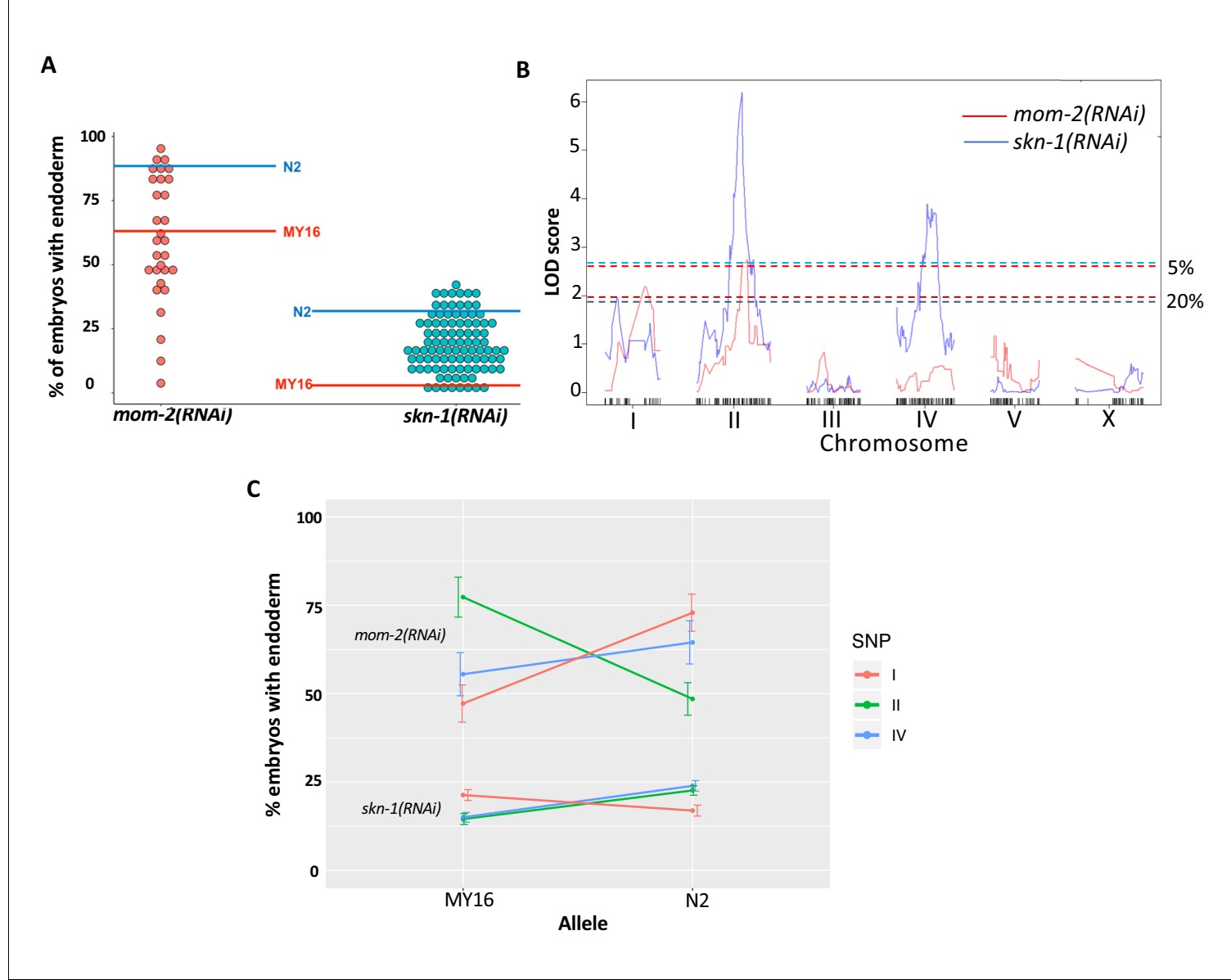

**Figure 6.** Quantitative genetic analysis of *mom-2(RNAi)* and *skn-1(RNAi)* phenotype in Recombinant Inbred Lines (RILs) between N2 and MY16. (**A**) *mom-2(RNAi)* (left) and *skn-1(RNAi)* (right) phenotype of RILs. The phenotype of the parental strains, MY16 and N2 are shown by red and blue lines, respectively. (**B**) QTL analyses (interval mapping) of *skn-1(RNAi)* (blue line) and *mom-2(RNAi)* (red line) phenotype shown in (**A**). Genomic regions are shown on the x-axis and LOD score is shown on the y-axis. Significance thresholds for *mom-2(RNAi)* and *skn-1(RNAi)* at 5% and 20% linkage represented in red and blue dashed lines, respectively. (**C**) Effect plots of significant SNPs from *mom-2(RNAi)* and *skn-1(RNAi)*, indicated by chromosome number and color, showing the direction of the allelic effects. Confidence intervals for the average phenotype in each genotype group are shown.

DOI: https://doi.org/10.7554/eLife.48220.014

## Potential cryptic relationships between SKN-1 and MOM-2 inputs

The preceding findings unveiled wide cryptic variation in the requirements for both SKN-1 and MOM-2/Wnt in the endoderm GRN, raising the possibility that the variation affecting the two inputs might be related. Indeed, comparisons of the GWAS and QTL mapping results for *skn-1* and *mom-2* showed an overlap in candidate QTL regions on chromosome I, II and IV (*Figure 5*, *Figure 6*, *Figure 5—figure supplement 2*), suggesting a possible connection between the genetic basis underlying these two traits. It is conceivable that some genetic backgrounds are generally more sensitive to

loss of either input (e.g., the threshold for activating the GRN is higher) and others more robust to single-input loss. Alternatively, a higher requirement for one input might be associated with a relaxed requirement for the other, that is, a reciprocal relationship.

As an initial assessment of these alternatives, we examined whether the requirements for SKN-1 and MOM-2 across the strains were significantly correlated. This analysis revealed no strong relationship between the cryptic variation in the requirement for these inputs seen across all the strains (Spearman correlation R = 0.18, p-value=0.07) (*Figure 7A*). This apparent lack of correlation at the level of strains is not unexpected, as many factors likely contribute to the cryptic variation and the comparison reflects the collective effect of all causal loci in the genome of each strain (*Figures 5* and *6*).

We next sought to examine possible relationships between the two GRN inputs at higher resolution by comparing association of specific genetic regions with the quantitative requirement for each input. We used the available sequencing data for all isotypes tested (*Andersen et al., 2012*) and examined the impact of each allele on the *skn-1(RNAi)* and *mom-2(RNAi)* phenotypes, correcting for outliers and using a pruned SNP map (see Materials and methods). We found a weak positive correlation (Pearson's R = 0.21, p=p value<2.2e-16, *Figure 7—figure supplement 1*) between the allelic effects. One possible explanation for this observation might be that variants across the set of wild isolates may generally influence the threshold for activating the positive feedback loops that lock down gut development (*Raj et al., 2010*; *Sommermann et al., 2010*), thereby altering the sensitivity for regulatory inputs into the endoderm pathway. Alternatively, although evidence for variation in germline RNAi sensitivity among *C. elegans* wild isolates is lacking (except for CB4856, which has been omitted from our study) (*Tijsterman et al., 2002*; *Félix et al., 2011*), and we have shown above that variation in SKN-1 and MOM-2 requirement reflects in large part cryptic genetic differences in the endoderm GRN, it remains possible that a fraction of the variation found in the two phenotypes tested is attributable to varying RNAi penetrance, which may underlie the minor positive correlation between *skn-1(RNAi)* and *mom-2(RNAi)* phenotypes.

In contrast, analysis of the N2/MY16 RILs uncovered a potential reciprocal relationship between the requirements for SKN-1 and MOM-2: we observed a negative correlation between the *skn-1 (RNAi)* and *mom-2(RNAi)* phenotypes across the genome (*Figure 7B*, genome-wide Pearson's R = −0.35, p=0.001, correcting for LD and outliers as above; correlation without chromosome IV R = −0.59, p<0.001) (*Figure 7B*). This finding suggested that at least some quantitative variants result in opposing effects on the requirement for SKN-1 and MOM-2.

While a reciprocal relationship was observed generally across the genome spanning five of the chromosomes, we observed the opposite correlation on chromosome IV (Pearson's R = 0.83, p-value=1.695×10$^{-5}$). No correlation was observed for chromosome IV with the wild isolates (Pearson's R = 0.08, NS, *Figure 7—figure supplement 1*). As there is a major QTL on chromosome IV for the SKN-1 requirement and there is substantial overlap in the same region with the GWAS analysis of the *skn-1(RNAi)* and *mom-2(RNAi)* phenotypes, we sought to dissect further the relationship between the requirement for MOM-2 and SKN-1 in this region. We created six near-isogenic lines (NILs) in which the QTL region for the *skn-1(RNAi)* phenotype on chromosome IV from N2 was introgressed into the MY16 background, and *vice-versa* (*Figure 7—figure supplement 2*). Control lines were created from the same crosses at the same generation by selecting the original parental region (e.g., selecting for the N2 region in an N2 background and MY16 in MY16 background). We found that the region affects the *skn-1(RNAi)* phenotype as expected: the N2 region increased the fraction of gut in an MY16 background, and the MY16 regions decreased this fraction in an N2 background. However, for *mom-2(RNAi)*, while introgressing the N2 region in MY16 dramatically changed the phenotype (*Figure 7C*), we found that the MY16 region was not sufficient to alter the phenotype in an N2 background. We created segregant NILs in which one of the genetic markers was lost (see Materials and methods) and found that replacing the N2 region with the corresponding MY16 region in all cases results in a stronger *mom-2(RNAi)* phenotype. However, for the *skn-1(RNAi)* phenotype six of nine segregants showed the opposite effect: that is, a weaker phenotype (*Figure 7D*), revealing that when contributing variants were separated by recombination, a reciprocal effect was frequently seen. These observations suggest that complex genetic interactions between variants on chromosome IV might mask the potential reciprocal effects that were observed on the other chromosomes.

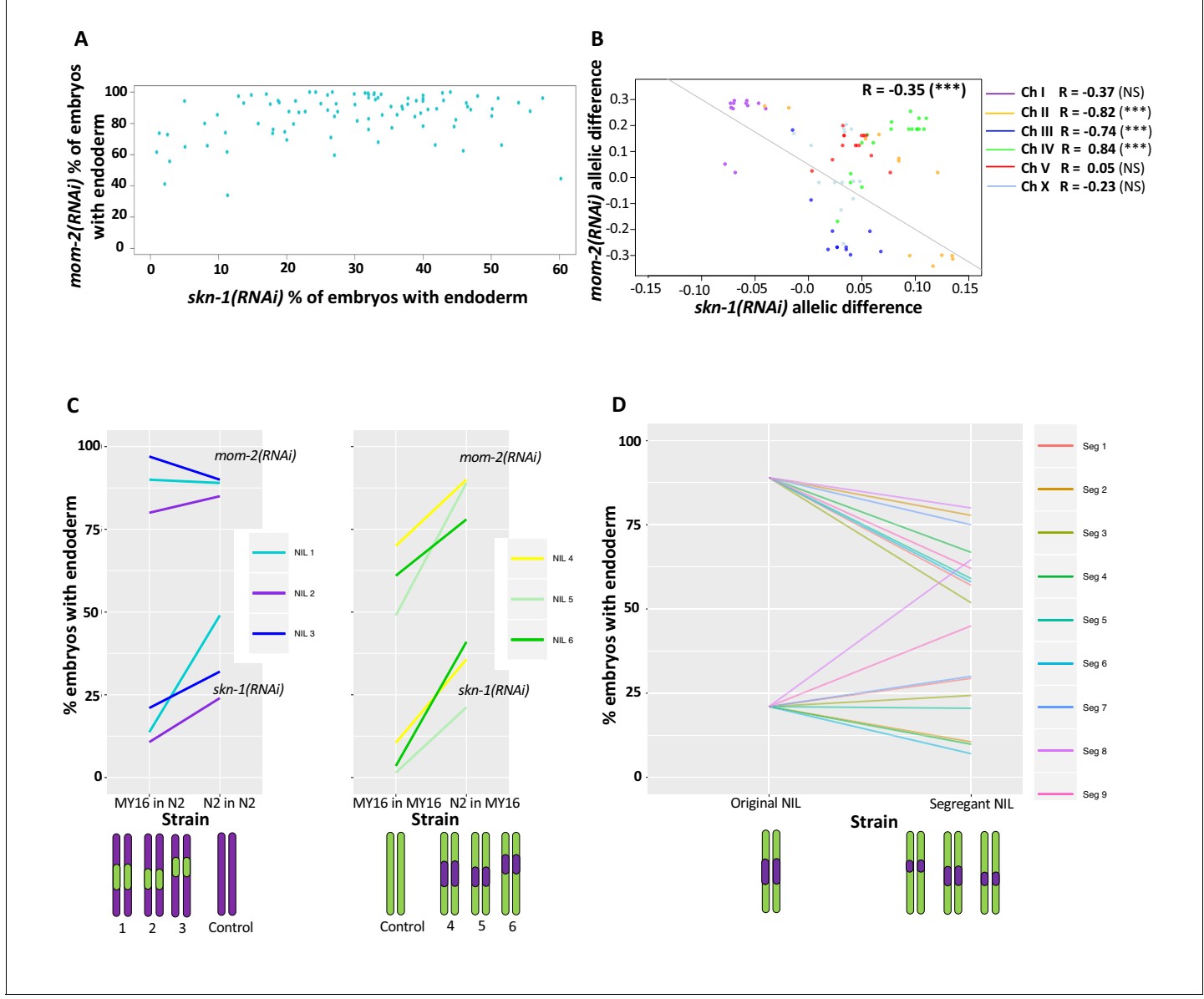

**Figure 7.** Correlation of *skn-1(RNAi)* and *mom-2(RNAi)* allelic differences. (**A**) Comparison of *skn-1(RNAi)* and *mom-2(RNAi)* phenotype in 94 natural isolates tested. No correlation was found (Spearman correlation R = 0.1844, p-value=0.07). Each dot corresponds to a wild isolate. Y-axis, *skn-1(RNAi)* phenotype, x-axis, *mom-2(RNAi)* phenotype. (**B**) Genome-wide correlation of *skn-1(RNAi)* and *mom-2(RNAi)* allelic differences in the N2xMY16 RILs. Each dot represents a SNP. Chromosomes are color-coded with their Pearson's R values represented (NS = Not Significant, ***=p value<0.001). Regression line in gray. (**C**) Six different NILs were created for chromosome IV, each of which was compared with a control NIL from the same cross (e. g., MY16 in MY16 as control for N2 in MY16). A schematic of the introgressed regions is represented below the plots. Percentage of *skn-1(RNAi)* or *mom-2(RNAi)* embryos with gut is represented. (**D**) Changes in phenotype for both *skn-1(RNAi)* and *mom-2(RNAi)* following recombination in segregant NILs, which a schematic representation of the segregant NILs below the plot.

DOI: https://doi.org/10.7554/eLife.48220.015

The following figure supplements are available for figure 7:

**Figure supplement 1.** Correlation between the *skn-1(RNAi)* and *mom-2(RNAi)* allelic differences.
DOI: https://doi.org/10.7554/eLife.48220.016

**Figure supplement 2.** Introgression of N2 and MY16 regions in chromosome IV.
DOI: https://doi.org/10.7554/eLife.48220.017

## Multiple factors reciprocally regulate the requirement for SKN-1 and MOM-2/Wnt

While the above findings revealed that the relationship between the requirement for SKN-1 and MOM-2 may be complicated by genetic interactions, our results raised the possibility of compensatory relationships between them. To further assess this possibility, we tested other candidate genes that reside in the QTL regions and that have been implicated in endoderm development (*Ruf et al., 2013*; *Witze et al., 2009*; *Walston et al., 2004*). We found that loss of RICT-1, the *C. elegans* orthologue of the human RICTOR (Rapamycin-insensitive companion of mTOR; *Tatebe and Shiozaki, 2017*), a component of the TORC2 complex, which has been shown to antagonize SKN-1 function (*Ruf et al., 2013*), results in opposite effects on *skn-1(-)* and *mom-2(-)* mutants (*Figure 8A*). Specifically, while *rict-1(RNAi)* suppresses the absence of gut in *skn-1(zu67)* embryos (*skn-1(zu67)*: 34.3% ± s.d 4.1% with gut *vs. skn-1(zu67); rict-1(RNAi)*: 48.3% ± s.d 4.9%; p=<0.001), we found that it *enhances* this phenotype in *mom-2(or42)* mutants (*mom-2(or42)*: 23.8% ± s.d 2.0%; vs. *mom-2 (or42); rict-1(RNAi)*: 11.2% ± s.d 3.2%; p<0.001). Confirming this effect, a similar outcome was observed when SKN-1 was depleted by RNAi in *rict-1(ft7)* chromosomal mutants (*skn-1(RNAi)*: 31.6% ± s.d 4.3% with gut vs. *rict-1(ft7); skn-1(RNAi)*: 45.9% ± s.d 6.3%; p<0.05) (*Figure 8A*). Similarly, RNAi depletion of PLP-1, the *C. elegans* homologue of the Pur alpha transcription factor that has been shown to bind to and regulate the *end-1* promoter (*Witze et al., 2009*), reciprocally affects the outcome of removing these two inputs in the same direction: loss of PLP-1 function suppresses the *skn-1(-)* phenotype (to 48.0% ± s.d 6.6%), and strongly enhances the *mom-2* phenotype (to 6.9% ± s.d 1.6%). Again, this result was confirmed by RNAi of *skn-1* in a *plp-1(ok2156)* chromosomal mutant (*Figure 8B*). Thus, as observed with the effect across the genome with natural variants, we observed a substantial reciprocal effect of both of these genes on loss of SKN-1 and MOM-2.

We also observed a reciprocal effect on the SKN-1 and Wnt inputs with MIG-5/*dishevelled*, a component of the Wnt pathway that acts downstream of the Wnt receptor (*Walston et al., 2004*); however, in this case the effect was in the opposite direction as seen for RICT-1 and PLP-1. Loss of MIG-5 as a result of chromosomal mutation or RNAi leads to *enhancement* of the *skn-1(-)* phenotype (*mig-5(rh94); skn-1(RNAi)*: 6.6% ± s.d 2.3%; *skn-1(zu67); mig-5(RNAi)*: 9.4% ± s.d 1.4%) and *suppression* of the *mom-2(-)* phenotype (88.6% ± s.d 4.0%) (*Figure 8C*).

Together, these findings reveal that, as observed with many of the N2/MY16 RILs variants across most of the genome, RICT-1, PLP-1, and MIG-5 show opposite effects on the phenotype of removing SKN-1 and MOM-2, suggesting a trend toward genetic influences that reciprocally influence the outcome in the absence of these two inputs.

## Discussion

The remarkable variety of forms associated with the ~36 animal phyla (*Adoutte and Philippe, 1993*) that emerged from a common metazoan ancestor >600 Mya is the product of numerous incremental changes in GRNs underlying the formation of the body plan and cell types (*Peter and Davidson, 2011*; *Carroll, 2008*). Here, we describe an unexpectedly broad divergence in the deployment of SKN-1/Nrf and MOM-2/Wnt signaling in generating the most ancient germ layer, the endoderm, within wild isolates of a single animal species, *C. elegans*. In this study, we report five major findings: 1) while the quantitative requirement for two distinct regulatory inputs that initiate expression of the endoderm GRN (SKN-1 and MOM-2) are highly reproducible in individual *C. elegans* isolates, there is wide cryptic variation between isolates. 2) Cryptic variation in the requirement for these regulatory factors shows substantial differences even between closely related strains, suggesting that these traits are subject to rapid evolutionary change in this species. 3) Quantitative genetic analyses of natural and recombinant populations revealed multiple loci underlying the variation in the requirement for SKN-1 and MOM-2 in endoderm specification. 4) The requirements for SKN-1 and MOM-2 in endoderm specification is frequently reciprocal in their relation to other genetic factors. 5) *rict-1, plp-1, and mig-5* reciprocally influence the outcome of *skn-1(-)* and *mom-2(-)*, substantiating the reciprocal influences on the two GRN inputs. These findings reveal prevalent plasticity and complexity underlying SKN-1 and MOM-2/Wnt regulatory inputs in mobilizing a conserved system for endoderm specification. Thus, while the core genetic toolkit for the development of the endoderm, the most ancient of the three germ layers, appears to have been preserved for well over half a billion

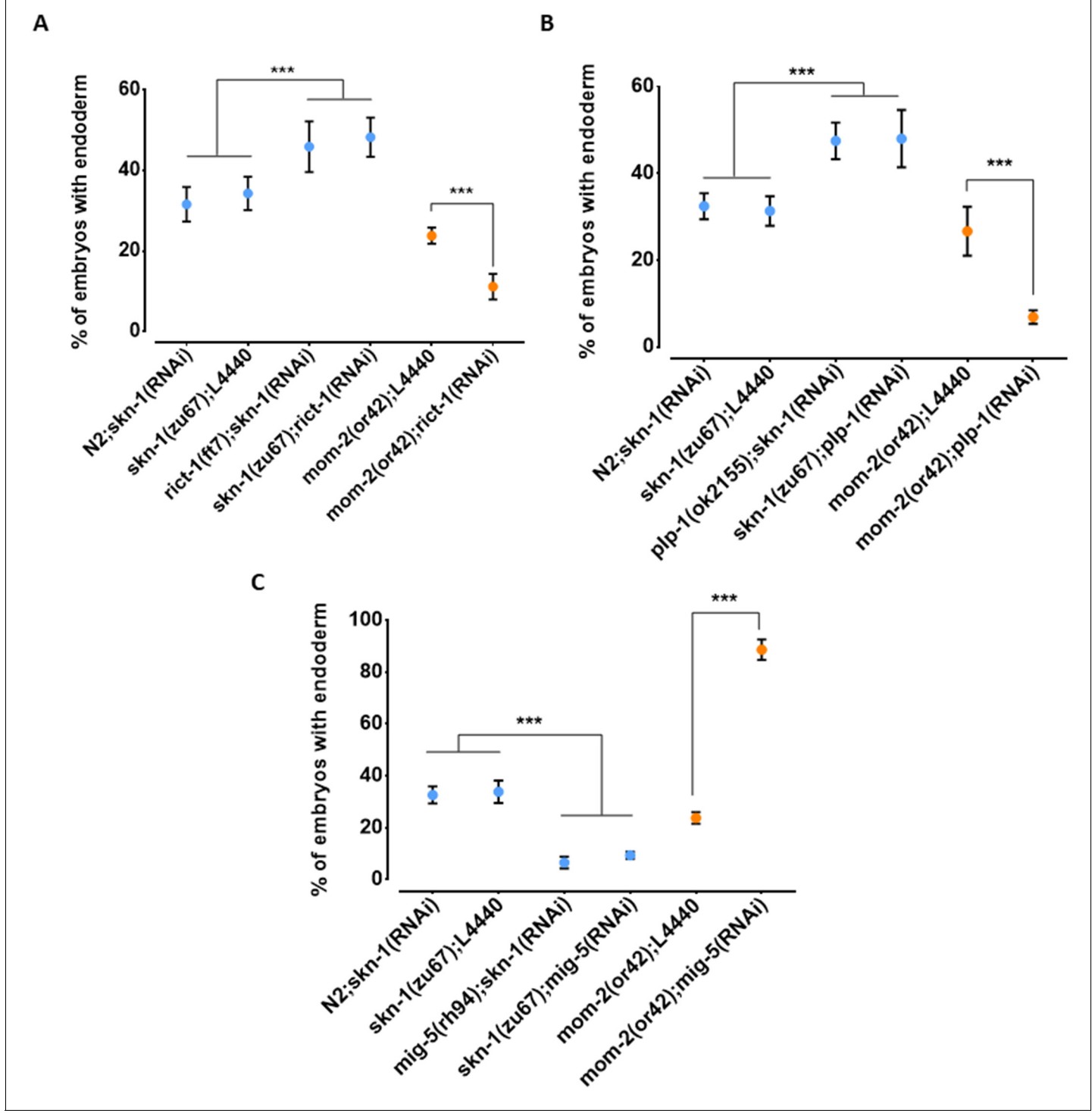

**Figure 8.** Reciprocal effects of RICT-1, PLP-1, and MIG-5 on *skn-1(-)* and *mom-2(-)* phenotypes. (A, B) Loss of RICT-1 or PLP-1 enhances the *mom-2 (or42)* loss-of-endoderm phenotype and suppresses *skn-1(zu67)* and *skn-1(RNAi)* phenotype. (C) Loss of MIG-5 enhances the *skn-1(zu67)* and *skn-1(RNAi)* phenotype and suppresses *mom-2(or42)* phenotype. At least three replicates were performed per experiment and >200 embryos per experiment. Student t-test (*** p-value<0.001). Error bars represent standard deviations.

DOI: https://doi.org/10.7554/eLife.48220.018

years, the molecular regulatory inputs that initiate its expression in *C. elegans* vary extremely rapidly over short evolutionary time scales within the species.

## Multigenic variation in the requirement for SKN-1 and MOM-2

Quantitative analyses of the wild isolates and RILs revealed that multigenic factors are responsible for the difference in requirement for SKN-1 and MOM-2 between isotypes. Notably, we observed substantial overlap on chromosome IV in the GWAS analyses of the *skn-1* and *mom-2* requirements in wild isotypes (*Figure 5*, *Figure 5—figure supplement 2* ) and on chromosome II from analyses using RILs (*Figure 6B*). This finding raises the possibility that some QTLs may influence requirement for both inputs into the endoderm specification pathway: as SKN-1 and Wnt converge to regulate expression of the *end-1/3* genes, it is conceivable that common genetic variants might modulate the relative strength or outcome of both maternal inputs. However, our findings do not resolve whether these genetic variants act independently to influence the maternal regulatory inputs. Genetic interactions are often neglected in large-scale genetic association studies (*Jenkins et al., 2009*) owing in part to the difficulty in confirming them (*Page et al., 2003*). Many studies (*Mackay, 2014*; *Volis et al., 2011*; *Félix, 2007*; *Barkoulas et al., 2013*), including ours here, showed that epistasis can strongly influence the behavior of certain variants upon genetic perturbation. In addition, selection on pleiotropically acting loci facilitates rapid developmental system drift. Together, epistasis and selection on pleiotropic loci play important roles in the evolution of natural populations (*Duveau and Félix, 2012*; *Johnson and Porter, 2007*; *Phillips, 2008*; *Wagner and Zhang, 2011*).

## Potential compensatory relationships between SKN-1 and MOM-2/Wnt

Although we did not observe a direct correlation between the *skn-1(-)* and *mom-2(-)* phenotypes across the isotypes studied here, we found a negative correlation across much of the genome for the N2 X MY16 RILs (*Figures 6* and *7*). Further, while GWAS and QTL analysis of natural and inbred lines, respectively, did not reveal a causal region in chromosome IV for *mom-2(RNAi)* variation, analysis of NILs results did uncover at least one QTL affecting this phenotype. Moreover, while broad regions of the chromosome showed a positive correlation between the SKN-1 and MOM-2 requirements, isolation of variants in NILs revealed an inverse requirement for these inputs for at least some regions on this chromosome. These results reflect the limitations of genome-wide studies of complex genetic traits: in the case of chromosome IV, several closely linked loci appear to influence both the SKN-1 and MOM-2 requirements.

Our findings raise the possibility that the SKN-1 and MOM-2/Wnt inputs might compensate for each other and that genetic variants that enhance the requirement for one of the inputs may often relax the requirement for the other. Such reciprocality could reflect cross-regulatory interactions between these two maternal inputs or could be the result of evolutionary constraints imposed by selection on these genes, which act pleiotropically in a variety of processes. Further supporting this possibility, we identified two genes, *rict-1* and *plp-1*, that show similar inverse effects on the requirements from *skn-1* and *mom-2*: debilitation of either gene enhances the phenotype of *mom-2(-)* and suppresses that of *skn-1(-)*. RICT-1 function extends lifespan in *C. elegans* through the action of SKN-1 (*Ruf et al., 2013*), and loss of RICT-1 rescues the misspecification of the MS and E blastomeres and lethality of *skn-1(-)* embryos (*Ruf et al., 2013*), consistent with our finding. We previously reported that PLP-1, a homologue of the vertebrate transcription factor pur alpha, binds to the *end-1* promoter and acts in parallel to the Wnt pathway and downstream of the MAPK signal (*Witze et al., 2009*), thereby promoting gut formation. PLP-1 shows a similar reciprocal relationship with SKN-1 and MOM-2 as with RICT-1 (*Figure 8*). Given that PLP-1 binds at a *cis* regulatory site in *end-1* near a putative POP-1 binding site (*Witze et al., 2009*), and that SKN-1 also binds to the *end-1* regulatory region (*Zhu et al., 1997*), it is conceivable that this reciprocality might reflect integration of information at the level of transcription factor binding sites. As the architecture of the GRN is shaped by changes in cis-regulatory sequences (*Peter and Davidson, 2011*; *Davidson and Levine, 2008*), analyzing alterations in SKN-1 and Wnt/POP-1 targets among *C. elegans* wild isolates may provide insights into how genetic changes are accommodated without compromising the developmental output at microevolutionary time scale.

MIG-5, a *dishevelled* orthologue, functions in the Wnt pathway in parallel to Src signaling to regulate asymmetric cell division and endoderm induction (*Bei et al., 2002*; *Walston et al., 2004*). We

found that the loss of *mig-5* function enhances the gut defect of *skn-1(-)* and suppresses that of the *mom-2(-), the* opposite reciprocal relationship to that of *rict-1* and *plp-1*, and consistent with a previous report (*Figure 8*) (*Bei et al., 2002*). These effects were not observed in embryos lacking function of *dsh-2*, the orthologue of *mig-5* (data not shown), supporting a previous study that showed overlapping but non-redundant roles of MIG-5 and DSH-2 in EMS spindle orientation and gut specification (*Walston et al., 2004*). Recent studies showed that Dishevelled can play both positive and negative roles during axon guidance (*Shafer et al., 2011*; *Zheng et al., 2015*). Dishevelled, upon Wnt-activation, promotes hyperphosphorylation and inactivation of Frizzled receptor to fine-tune Wnt activity. It is tempting to speculate that MIG-5 may perform similar function in EMS by downregulating activating signals (Src or MAPK), in the absence of MOM-2.

We hypothesize that compensatory mechanisms might evolve to fine-tune the level of gut-activating regulatory inputs. Successful developmental events depend on tight spatial and temporal regulation of gene expression. For example, anterior-posterior patterning in the *Drosophila* embryo is determined by the local concentrations of the Bicoid, Hunchback, and Caudal transcription factors (*Rivera-Pomar and Jäckle, 1996*). We postulate that SKN-1 and Wnt signaling is modulated so that the downstream genes, *end-1/3*, which control specification and later differentiation of endoderm progenitors, are expressed at optimal levels that ensure normal gut development. Suboptimal END activity leads to poorly differentiated gut and both hypo- and hyperplasia in the gut lineage (*Maduro et al., 2015*; *Choi et al., 2017*; *Maduro, 2015*). Hyper- or hypo-activation of Wnt signaling has been implicated in cancer development (*Zhan et al., 2017*), bone diseases (*Jenkins et al., 2009*; *Baron and Gori, 2018*), and metabolic diseases (*Chen and Wang, 2018*; *Schinner, 2009*), demonstrating the importance of regulating the timing and dynamics of such developmental signals within a quantitatively restricted window.

## Cryptic variation and evolvability of GRNs

This study revealed substantial cryptic genetic modifications that alter the relative importance of two partially redundant inputs into the *C. elegans* endoderm GRN, leading to rapid change in the developmental network architecture (*Figure 9*). Such modifications may occur through transitional states that are apparent even within this single species. For example, the finding that POP-1 is not required for gut development even in a wild isolate (e.g., MY16) that, like *C. briggsae*, shows a near-absolute requirement for SKN-1 may reflect a transitional state between the two species: that is, a nearly essential requirement for SKN-1 but non-essential requirement for POP-1, an effect not previously seen in either species. In addition, duplicated GATA factors (the MEDs, ENDs, and ELTs) and partially redundant activating inputs (SKN-1, Wnt, Src, and MAPK) in endoderm GRN, provide an opportunity for genetic variation to accumulate and 'experimentation' of new regulatory relationships without diminishing fitness (*Félix and Wagner, 2008*; *Schinner, 2009*; *Frankel et al., 2010*).

Redundancy in the regulatory inputs may act to 'rescue' an initial mutation and allow for secondary mutations that might eventually lead to rewiring of the network. For example, loss of either MyoD or Myf5, two key regulators of muscle differentiation in metazoans, produces minimal defects in myogenesis as a result of compensatory relationship between the myogenic factors (*Mohun, 1992*). In vertebrates, gene duplication events have resulted in an expansion of Hox genes to a total of >200, resulting in prevalent redundancy (*Imai et al., 2001*; *Manley and Capecchi, 1997*; *Nam and Nei, 2005*). This proliferation of redundant genes provides opportunities for evolutionary experimentation, subsequent specialization of new functions, and developmental system drift (*Nam and Nei, 2005*; *True and Haag, 2001*). In *C. elegans*, loss of GAP-1 (a Ras inhibitor) or SLI-1 (a negative regulator of EGFR signaling) alone does not produce obvious defects, while double mutations lead to a multivulva phenotype (*Yoon et al., 2000*). Similar redundant relationships between redundant partners exist in many other contexts in the animal. Notably, the relative importance of Ras, Notch, and Wnt signals in vulva induction differ in various genetic backgrounds (*Milloz et al., 2008*; *Gleason et al., 2002*) and physiological conditions (*Braendle and Félix, 2008*; *Grimbert et al., 2018*), resulting in flexibility in the system. While vulval development in *C. elegans*, when grown under standard laboratory conditions, predominantly favors utilization of the EGF/Ras signaling pathway (*Braendle and Félix, 2008*), Wnt is the predominant signaling pathway in the related *Pristionchus pacificus*, which is ~250 MY divergent (*Zheng et al., 2005*; *Tian et al., 2008*). In addition, while *Cel-lin-17* functions positively to transduce the Wnt signal, *Ppa-lin-17/Fz* antagonizes Wnt signaling and instead the Wnt signal is transmitted by *Ppa-lin-18/Ryk*, which has acquired a

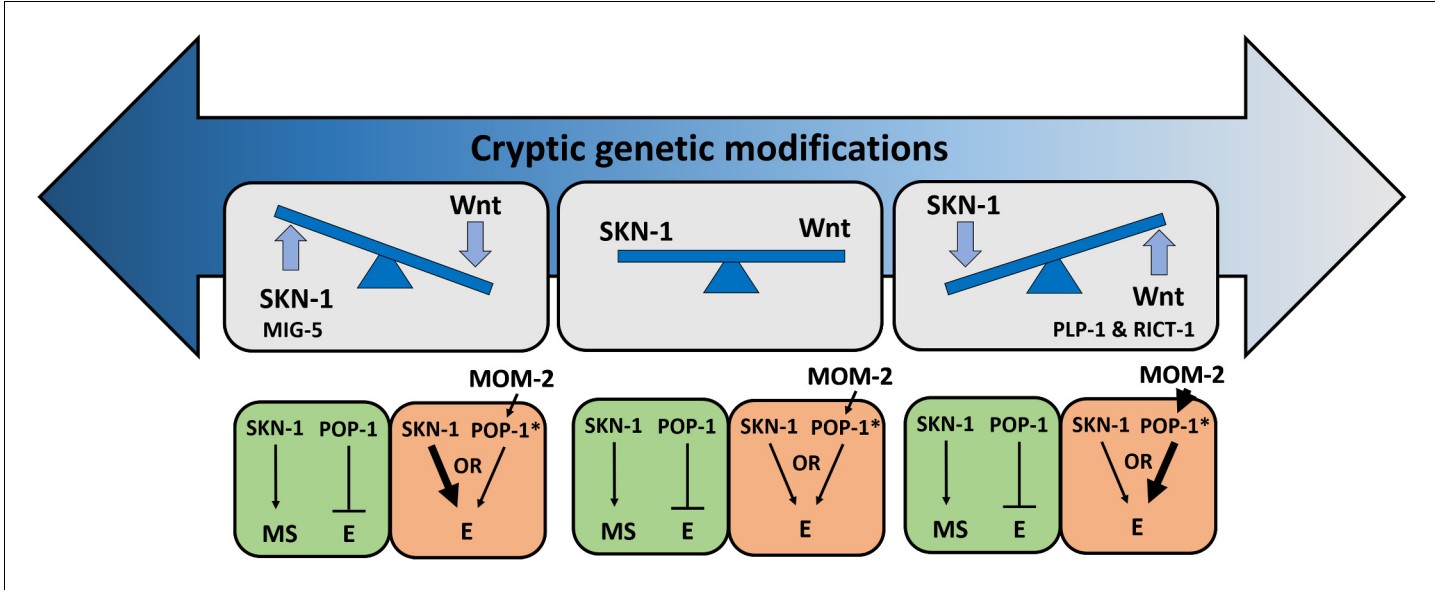

**Figure 9.** Simplified models for potential cryptic compensatory relationship between the SKN-1 and MOM-2/Wnt regulatory inputs in the endoderm GRN. Accumulation of cryptic genetic modifications drives rapid rewiring of the GRN, causing broad variation of SKN-1 and MOM-2/Wnt dependence in endoderm (E) specification among *C. elegans* isotypes. Wnt-signaled POP-1 (indicated by *) acts as an E activator, while unmodified POP-1 in the MS blastomere acts as a repressor of E fate in all *C. elegans* isotypes. The relative strength of the inputs is indicated by the thickness of the arrow. RICT-1, PLP-1 and MIG-5 reciprocally influence the outcome in the absence of the two inputs.
DOI: https://doi.org/10.7554/eLife.48220.019

novel SH3 domain not present in the *C. elegans* ortholog (*Wang and Sommer, 2011*). Thus, extensive rewiring of signaling networks and modularity of signaling motifs contribute to developmental systems drift (*True and Haag, 2001*; *Haag et al., 2018*).

The broad cryptic variation may drive developmental system drift, giving rise to GRN architectures that differ in the relative strength of the network components. Our finding that the key regulatory inputs that initiate the endoderm GRN show dramatic plasticity is consistent with comparative transcriptomic studies that demonstrate high gene expression variability and divergence during early embryonic stages in fly (*Kalinka et al., 2010*; *Gerstein et al., 2014*; *Levin et al., 2016*), worm (*Gerstein et al., 2014*; *Levin et al., 2016*; *Levin et al., 2012*; *Zalts and Yanai, 2017*), Xenopus (*Levin et al., 2016*; *Yanai et al., 2011*; *Irie and Kuratani, 2011*), zebrafish (*Levin et al., 2016*; *Irie and Kuratani, 2011*), and mouse (*Levin et al., 2016*; *Irie and Kuratani, 2011*). Therefore, early developmental events may be highly evolvable, in part due to weak purifying selection on maternal-effect genes (*Cruickshank and Wade, 2008*; *Cutter et al., 2019*). This is in accordance with the 'hourglass' concept of embryonic development (*Kalinka et al., 2010*; *Raff, 1996*; *Domazet-Lošo and Tautz, 2010*), in which divergent developmental mechanisms during early embryogenesis converge on a more constant state (i.e., a 'phylotypic stage' at the molecular regulatory level). Indeed, unlike the terminal differentiation factor ELT-2, upstream MEDs and ENDs genes are present only in closely related Caenorhabditis species (*Gillis et al., 2007*; *Maduro, 2015*; *Coroian et al., 2006*; *Maduro et al., 2005b*). This is likely attributable either to positive selection during early embryonic and later larval stages or to developmental constraints. Analysis of developmental gene expression in mutation accumulation lines, which have evolved in the absence of any positive selection, showed similarity to the developmental hourglass model of evolvability, consistent with strong developmental constraints on the phylotypic stage (*Zalts and Yanai, 2017*). However, they do not rule out the possibility that early and late stages of development might be more adaptive and therefore subject to positive selection. It will be of interest to learn the degree to which the divergence in network architecture might arise as a result of differences in the environment and selective pressures on different *C. elegans* isotypes.

# Materials and methods

## Key resources table

| Reagent type (species) or resource | Designation | Source or reference | Identifiers | Additional information |
|---|---|---|---|---|
| Strain, strain background (*C. elegans*) | Wild isolates; refer to *Supplementary file 1* | CGC | | |
| Strain, strain background (*C. elegans*) | JJ185 | CGC | | dpy-13(e184) skn-1(zu67) IV; mDp1 (IV;f). |
| Strain, strain background (*C. elegans*) | JR3666 | This study | | (elt-2::GFP) X; (ifb-2::CFP) IV; see *Figure 1* |
| Strain, strain background (*C. elegans*) | EU384 | CGC | | dpy-11(e1180) mom-2(or42) V/nT1 [let-?(m435)] (IV;V). |
| Strain, strain background (*C. elegans*) | JJ1057 | CGC | | pop-1(zu189) dpy-5(e61)/hT1 I; him-5(e1490)/hT1 V. |
| Strain, strain background (*C. elegans*) | KQ1366 | CGC | | rict-1(ft7) II. |
| Strain, strain background (*C. elegans*) | SU351 | CGC | | mig-5(rh94)/mIn1 [dpy-10(e128) mIs14] II. |
| Strain, strain background (*C. elegans*) | RB1711 | CGC | | plp-1(ok2155) IV. |
| Strain, strain background (*C. elegans*) | JR3493-JR3590 (refer to *Supplementary file 2*) | This study | | N2xMY16 RILs; see *Figure 6* |
| Strain, strain background (*C. elegans*) | MT3414 | CGC | | dpy-20(e1282) unc-31(e169) unc-26(e205) IV. |
| Strain, strain background (*C. elegans*) | DA491 | CGC | | dpy-20(e1282) unc-30(e191) IV. |
| Strain, strain background (*C. elegans*) | JR2750 | *Kontani et al., 2005* | | bli-6(Sc16)unc-22(e66)/unc-24 (e138)fus-1(w13) dpy-20 (e2017)IV |
| Strain, strain background (*C. elegans*) | JR3812 (NIL 1) | This study | | NIL N2XMY16; see *Figure 7* |
| Strain, strain background (*C. elegans*) | JR3813 (NIL 2) | This study | | NIL N2XMY16; see *Figure 7* |
| Strain, strain background (*C. elegans*) | JR3814 (NIL 3) | This study | | NIL N2XMY16; see *Figure 7* |
| Strain, strain background (*C. elegans*) | JR3815 (NIL 4) | This study | | NIL N2XMY16; see *Figure 7* |
| Strain, strain background (*C. elegans*) | JR3816 (NIL 5) | This study | | NIL N2XMY16; see *Figure 7* |
| Strain, strain background (*C. elegans*) | JR3817 (NIL 6) | This study | | NIL N2XMY16; see *Figure 7* |

*Continued on next page*

*Continued*

| Reagent type (species) or resource | Designation | Source or reference | Identifiers | Additional information |
|---|---|---|---|---|
| Antibody | MH33 mouse monoclonal | DSHB | RRID:AB_528311 | 1:50 dilution |
| Antibody | AHP418 rabbit polyclonal | Serotec Bio-Rad | RRID:AB_2116715; PMID:28736134 | 1:200 dilution |
| Antibody | ab150116 goat polyclonal | Abcam | RRID:AB_2650601; PMID:31167447 | Goat Anti-Mouse IgG H and L (Alexa Fluor 594) |
| Antibody | ab150077 goat polyclonal | Abcam | RRID:AB_2630356 | Goat Anti-Rabbit IgG H and L (Alexa Fluor 488) |
| Software, algorithm | R v 3.2.3 | The R Foundation | RRID:SCR_001905 | |
| Software, algorithm | PLINK | http://pngu.mgh.harvard.edu/purcell/plink/ | RRID:SCR_001757 | |

## C. elegans strains and maintenance

All wild isolates, each with a unique haplotype (*Andersen et al., 2012*), were obtained from the Caenorhabditis Genetics Center (CGC) (see *Supplementary file 1*). Worm strains were maintained as described (*Brenner, 1974*) and all experiments were performed at 20°C unless noted otherwise.

## RNAi

Feeding-based RNAi experiments were performed as described (*Kamath and Ahringer, 2003*). RNAi clones were obtained from either the Vidal (*Rual et al., 2004*) or Ahringer libraries (*Kamath et al., 2003*). RNAi bacterial strains were grown at 37°C in LB containing 50 µg/ml ampicillin. The overnight culture was then diluted 1:10. After 4 hr of incubation at 37°C, 1 mM of IPTG was added and 60 µl was seeded onto 35 mm agar plates containing 1 mM IPTG and 25 µg/ml carbenicillin. Seeded plates were allowed to dry and used within five days. Five to 10 L4 animals were placed on RNAi plate. 24 hr later, they were transferred to another RNAi plate and allowed to lay eggs for four or 12 hr (12 hr for *skn-1* RNAi and four hours for the other RNAi). The adults were then removed, leaving the embryos to develop for an extra 7–9 hr. Embryos were quantified and imaged on an agar pad using a Nikon Ti-E inverted microscope. We chose to perform RNAi on agar plates to maximize sensitivity, robustness, and reproducibility of the assay, as liquid culture RNAi can introduce variability owing to aggregation and settling of bacteria, which affects RNAi efficacy (*Gomez-Amaro et al., 2015*). In addition, performing RNAi on agar plates allowed us to collect large numbers of embryos with which to quantify gut formation (as described below).

## Antibody staining

The embryonic gut cells and nuclei of all cells were stained with MH33 (mouse anti-IFB-2, deposited to the DSHB by Waterston, R.H.) and AHP418 (rabbit anti-acetylated histone H4, Serotec Bio-Rad) respectively. Fixation and permeabilization were carried out as described previously (*Sommermann et al., 2010*). Goat anti-mouse Alexa Fluor 594 and goat anti-rabbit Alexa Fluor 488 secondary antibodies were used at 1:1000 dilution.

## Quantification of endoderm specification

Gut was scored by the presence of birefringent gut granule in arrested embryos (*Clokey and Jacobson, 1986*; *Hermann et al., 2005*). For *skn-1(RNAi)*, the laboratory strain N2, which shows invariable ~30% of embryos with endoderm, was used as a control for all experiments.

## Introgression of *skn-1(zu67)*, *pop-1(zu189)*, and *mom-2(or42)* alleles into wild isolate backgrounds

To introgress *skn-1(zu67)* into wild isolates (WI), males from the wild isolate strains were crossed to JJ186 *dpy-13(e184) skn-1(zu67) IV; mDp1 (IV;f)* hermaphrodites. mDp1 is a free duplication

maintained extrachromosomally that rescues the Dpy and lethal phenotypes of *dpy-13(e184)* and *skn-1(zu67)* respectively. mDp1 segregates in a non-Mendelian fashion and animals that have lost the free duplication are Dpy and produce dead offspring. Wild type F1 hermaphrodites that have lost the free duplication, as determined by the presence of 1/4 Dpy progeny in the F2 generation, were selected. 10 single non-Dpy F2 hermaphrodite descendants from F1 animals heterozygous for *skn-1(zu67)* (2/3 of which are expected to be of the genotype *WI dpy-13(+) skn-1(+)/dpy-13(e184) skn-1(zu67)* were backcrossed to their respective parental wild strain. 10 F3 hermaphrodites were picked to individual plates. Half of the F3 cross progeny are expected to be heterozygous for *dpy-13(e184) skn-1(zu67)*, as evidenced by presence of F4 Dpy progeny that produced dead embryos. Non-Dpy siblings were used to continue the introgression as described. This strategy was repeated for at least five rounds of introgression. The embryonic gutless phenotype in the progeny of the Dpy animals was quantified.

Similarly, to introgress *pop-1(zu189)* or *mom-2(or42)* alleles into wild isolates, JJ1057 *pop-1 (zu189) dpy-5(e61)/hT1 I; him-5(e1490)/hT1V* or EU384 *dpy-11(e1180) mom-2(or42) V/nT1 [let-? (m435)] (IV;V)* were used, respectively. The mutant strain was crossed to the wild isolates. Non-Dpy F2 animals heterozygous for the chromosomal mutation were selected and backcrossed to their respective parental wild strain for at least four rounds of introgression for *pop-1* and seven rounds for *mom-2*. The embryonic gutless phenotype in the progeny of the Dpy animals was quantified, as above.

## Statistical analyses: GWAS

All data were analyzed and plotted using R software v 3.2.3 (https://www.r-project.org/). GWAS for both phenotypes was performed using *C. elegans* wild isolates and a previously published SNP map containing 4,690 SNPs (*Andersen et al., 2012*) with the EMMA R package. P-values were calculated using mixed model analysis (*Kang et al., 2008*) (emma.REML.t() function) and identity-by-state (IBS) kinship matrix to account for population structure. For *skn-1* and *mom-2* RNAi phenotypic data, a genome-wide permutation-based FDR was also calculated for the EMMA results from 10,000 permuted values (*Millstein and Volfson, 2013*; *Hansen and Kerr, 2012*).

## Phylogenetic and geographical analyses

Phylogenetic trees were constructed from 4690 polymorphisms using R package 'ape' (*Paradis et al., 2004*). Neighbor-joining algorithm based on pairwise distances was used. Phylogenetic signal (Pagel's λ statistics) was measured using 'phylosig()" function in phytools R package (*Revell, 2012*; *Ives et al., 2007*). Statistical significance of λ was obtained by comparing the likelihood a model accounting for the observed λ with the likelihood of a model that assumes complete phylogenetic independence.

Geographic information for strains were obtained from *Andersen et al. (2012)*, available in *Supplementary file 1*, together with the corresponding *skn-1(RNAi)* and *mom-2(RNAi)* phenotypes.

## Correlation analysis

To test for the relationship between *mom-2 (RNAi)* and *skn-1 (RNAi)* phenotypic data, the differences between median phenotypic values for each SNP were calculated independently on a genome-wide level for the wild isolates. In order to correct for LD, SNPs were pruned with PLINK (http://pngu.mgh.harvard.edu/purcell/plink/) (*Purcell et al., 2007*) and only a subset of SNPs was used for the correlation analyses. Outliers were removed from the calculations by using z-score with a cutoff of 1.96 (i.e., 95% of values fall within ±1.96 in a normal distribution).

## RIL construction and Genotype-By-Sequencing (GBS)

Recombinant inbred lines (RILs) were created by crossing an N2 hermaphrodite and an MY16 male. 120 F2 progeny were cloned to individual plates and allowed to self-fertilize for 10 generations. A single worm was isolated from each generation to create inbred lines. A total of 95 lines were successfully created and frozen stocks were immediately created and kept at −80℃ (*Supplementary file 2*), prior to DNA sequencing.

DNA was extracted using Blood and Tissue QIAGEN kit from worms from each of the RILs grown on four large NGM plates (90 × 15 mm) with OP50 *E. coli* until starved (no more than a day).

Samples were submitted in 96-well plate format at 10 ng/μl < n < 30 ng/μl. GBS libraries were constructed using digest products from ApeKI (GWCGC), using a protocol modified from *Elshire et al. (2011)*. After digestion, the barcoded adapters were ligated and fragments < 100 bp were sequenced as single-end reads using an Illumina HiSeq 2000 lane (100 bp, single-end reads).

SNP calling was performed using the GBSversion3 pipeline in Trait Analysis by aSSociation, Evolution and Linkage (TASSEL) (*Bradbury et al., 2007*). Briefly, fastq files were aligned to reference genome WS252 using BWA v. 0.7.8-r455 and SNPs were filtered using vcftools (*Danecek et al., 2011*). Samples with greater than 90% missing data and SNPs with minor allele frequencies (mAF) of <1% were excluded from analysis, identifying 27,396 variants.

## QTL mapping using R/qtl

Variants identified by GBS pipeline were filtered to match the SNPs present in the parental MY16 strain (using vcftools –recode command), and variants were converted to a 012 file (vcftools –012 command). Single-QTL analysis was performed in R/QTL (*Broman and Sen, 2009*) using 1770 variants and 95 RILs. Significant QTL were determined using Standard Interval Mapping (scanone() 'em') and genome-wide significance thresholds were calculated by permuting the phenotype (N = 1,000). Change in log-likelihood ratio score of 1.5 was used to calculate 95% confidence intervals and define QTL regions (*Broman et al., 2003*). SNP data for the RILs and their corresponding phenotypes used in analysis are shown in *Supplementary files 2* and *3*.

## Creation of near-isogenic lines (NILs)

Three N2-derived mutant strains were used to introgress regions from chromosome IV from N2 into the MY16 strain background and vice-versa. For both types of crosses, N2 was always used as the maternal line. The following strains were used:

- DA491: *dpy-20(e1282) unc-30(e191) IV.*
- JR2750: *bli-6(sc16) unc-22(e66)/unc-24(e138)fus-1(w13) dpy-20(e2017) IV*. Worms segregating Bli Unc were selected for crosses.
- MT3414: *dpy-20(e1282) unc-31(e169) unc-26(e205) IV.*

To introgress the N2 region into the MY16 genetic background, hermaphrodites from the N2-derived strains containing genetic markers flanking the genomic region of interest were crossed with MY16 males (*Figure 7—figure supplement 2*). After successful mating, 10 F1 heterozygotes were isolated and allowed to self. After 24 hr, the F1 adults were removed from the plate, and the F2 hermaphrodites left to develop to young adults. F2 animals homozygous for the region being introgressed were selected as young adults and crossed with MY16 males. This process was repeated until ten rounds of introgression were completed. These new lines were preserved at −80C.

Introgression of MY16 region into an N2 background began with the same initial cross as above. F1 heterozygous males were crossed with N2 hermaphrodites containing phenotypic markers near the region being introgressed (*Figure 7—figure supplement 2*). After successful mating, the F1 parents were removed and the F2 generation was left to develop until heterozygous males were visible. F2 heterozygous males were crossed with hermaphrodites from the N2-derived strain. This process was repeated until ten successful introgressions were completed. To homozygose the introgressed MY16 regions, worms were singled and allowed to self until a stable wildtype population was obtained. These new lines preserved at −80C.

NILs were genotyped to test for correct introgression of the desired regions by Sanger sequencing of 10 markers spaced along chromosome IV (carried out by the Centre for Genomics and Proteomics, University of Auckland). Upon confirmation of their genetic identity, one NIL was used to further dissect the QTL region by segregating the visual markers (*Dpy* and *Unc*).

## Acknowledgements

We thank members of the Rothman lab, especially Sagen Flowers and Kristoffer C Mellingen for experimental assistance, and Snell labs, particularly Dr. Kien Ly, for helpful advice and feedback. We thank Dr. Kathy Ruggiero (University of Auckland, New Zealand) for helpful advice on GWAS methodology and Dr. James McGhee (University of Calgary, Canada) for providing the MH33 antibody. Nematode strains used in this work were provided by the Caenorhabditis Genetics Center, which is

funded by the National Institutes of Health - Office of Research Infrastructure Programs (P40 OD010440). YNTC was supported during part of this work by a University of Auckland Doctoral Scholarship. This work was supported by grants from the NIH (#1R01HD082347 and # 1R01HD081266) to JHR.

## Additional information

### Funding

| Funder | Grant reference number | Author |
| --- | --- | --- |
| National Institutes of Health | 1R01HD082347 | Joel H Rothman |
| National Institutes of Health | 1R01HD081266 | Joel H Rothman |

The funders had no role in study design, data collection and interpretation, or the decision to submit the work for publication.

### Author contributions

Yamila N Torres Cleuren, Conceptualization, Resources, Data curation, Formal analysis, Supervision, Funding acquisition, Validation, Investigation, Visualization, Methodology, Writing—original draft, Writing—review and editing; Chee Kiang Ewe, Conceptualization, Resources, Data curation, Formal analysis, Funding acquisition, Validation, Investigation, Visualization, Methodology, Writing—original draft, Writing—review and editing; Kyle C Chipman, Conceptualization, Data curation, Investigation, Writing—original draft, Writing—review and editing; Emily R Mears, Conceptualization, Investigation, Writing—review and editing; Cricket G Wood, Coco Emma Alma Al-Alami, Investigation, Writing—review and editing; Melissa R Alcorn, Data curation, Investigation, Writing—review and editing; Thomas L Turner, Data curation, Writing—review and editing; Pradeep M Joshi, Conceptualization, Data curation, Supervision, Writing—review and editing; Russell G Snell, Conceptualization, Supervision, Writing—original draft, Writing—review and editing; Joel H Rothman, Conceptualization, Supervision, Funding acquisition, Methodology, Writing—original draft, Writing—review and editing

### Author ORCIDs

Yamila N Torres Cleuren https://orcid.org/0000-0003-2218-7243
Chee Kiang Ewe https://orcid.org/0000-0003-1973-1308
Pradeep M Joshi http://orcid.org/0000-0002-4220-0559
Joel H Rothman https://orcid.org/0000-0002-6844-1377

### Decision letter and Author response

Decision letter https://doi.org/10.7554/eLife.48220.028
Author response https://doi.org/10.7554/eLife.48220.029

## Additional files

### Supplementary files

• Supplementary file 1. Wild isolates and their corresponding skn-1(RNAi) and mom-2(RNAi) phenotype, along with isolation and genotype information.
DOI: https://doi.org/10.7554/eLife.48220.020

• Supplementary file 2. skn-1(RNAi) and mom-2(RNAi) phenotype of N2xMY16 RIL strains.
DOI: https://doi.org/10.7554/eLife.48220.021

• Supplementary file 3. Genotype data for each of the N2xMY16 RIL strains. Values of H are Heterozygotes, A is N2 allele, B is MY16 allele, missing allele data represented as "-".
DOI: https://doi.org/10.7554/eLife.48220.022

• Transparent reporting form
DOI: https://doi.org/10.7554/eLife.48220.023

## Data availability

All data generated or analysed during this study are included in the manuscript and supporting files.

The following previously published dataset was used:

| Author(s) | Year | Dataset title | Dataset URL | Database and Identifier |
|---|---|---|---|---|
| Andersen EC | 2012 | SNP calls | https://raw.githubusercontent.com/Andersen-Lab/Rad-Seq/master/data/41188.SNP.WS210.txt | Github, 047839 |

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
