## [Decision Letter]

Thank you for submitting your article "Reciprocal requirement of Wnt signalling and SKN-1 underlies cryptic intraspecies variation in an ancient embryonic GRN" for consideration by *eLife*. Your article has been reviewed by three peer reviewers, one of whom is a member of our Board of Reviewing Editors, and the evaluation has been overseen by Michael Eisen as the Senior Editor. The reviewers have opted to remain anonymous.

The reviewers have discussed the reviews with one another and the Reviewing Editor has drafted this decision to help you prepare a revised submission.

Summary:

This interesting manuscript by Torres-Cleuren et al. examines cryptic genetic variation among *C. elegans* wild isolates for the requirement of two key genes for gut specification in early embryos in the reference strain, *skn-1*/Nrf2 and *mom-2/Wnt*. When either gene is knocked down by RNAi in conditions resulting in 100% lethality, the various *C. elegans* natural isolates display dramatic variation in the penetrance of endoderm formation. Given possible caveats with RNAi efficency, the authors nicely provide additional support for the observed variation in penetrance in a few isolates by introgression of *skn-1* and *mom-2* mutations. A GWAS analysis shows association on chr IV for *skn-1(RNAi)* and no significant association for *mom-2(RNAi)*. The authors go on to build Recombinant Inbred Lines between two lines (N2 and MY16; MY16 displaying a lower penetrance than N2 for both genes) and at least find QTLs on chr IV and II for *skn-1(RNAi)*. A potentially interesting and particularly novel part of the study concerns the correlation between the phenotypes of *skn-1(RNAi)* versus *mom-2(RNAi)*. The authors do not detect correlation between the two phenotypes in the wild isolates (or a marginally significant positive one) yet find a negative genetic correlation across a large part of the genome when testing windows of 50 SNPs. Given this negative correlation, the authors then present experimental data showing that mutant/RNAi variation at three loci (*rict-1, mig-5, plp-1*) has reciprocal effects on *skn-1* and *mom-2 mutant/RNAi* effects. The authors suggest the existence of widespread compensatory cryptic genetic variation in the requirements for SKN-1 versus MOM-2. Cryptic genetic variation and compensatory evolution in developmental pathways are important topics in evolutionary developmental biology. The possible negative correlation between the requirement for the two genes is an important result, if confirmed.

All reviewers agreed that there are some major issues with the methods, particularly for the quantitative genetic analyses. The major concern is point 4 (correlation analysis) that conditions the title of the paper and thus requires further analyses.

Essential revisions:

1) Statements on RNAi sensitivity (e.g. third paragraph of subsection “Extensive natural cryptic variation in the requirement for SKN-1 in endoderm specification

within the *C. elegans* species”: "fully sensitive" to RNAi). Please make clear that sensitivity to RNAi is a quantitative trait. Although the authors place themselves in very strong RNAi conditions that result in 100% lethality (in contrast to a previous study by Paaby et al., 2015 where RNAi was weaker), they cannot conclude that RNAi efficiency is the same in all isolates or RILs. It may be that 0-5% of expression of the target gene causes 100% lethality, but a range of endoderm phenotypes. The introgression results are convincing for the tested isolates; however, it remains possible that some genetic variation for RNAi efficiency is detected. Of course, it would not explain the negative correlation between the two RNAi experiments, which may be worth to point out.

2) Comments on GWAS analyses:

– We suggest that the authors only show the EMMA results. Is there a reason to use a model that does not correct for population structure?

(NB: EMMA performs a GWAS, so do not refer to GWAS and EMMA as two different things.)

– Since the phenotypes are proportions, consider to use an arcsine transformation or best a binomial model on individual observations taking replicates into account. This may particularly help for the skewed *mom-2* data.

– Where does the EMMA peak on chromosome IV with *skn-1(RNAi)* map? Please provide the association results in a supplementary table. Since the *skn-1* gene is in this region (as far as can be judged), can you rule out that variation in RNAi efficiency due to *skn-1* polymorphism underlies it?

3) Regarding RILs:

– Is there any reason to perform marker regression versus interval mapping? Just go for interval mapping.

– Please comment whether all lines show 100% lethality upon RNAi of each gene.

– Figure 6A: why were so few lines tested for *mom-2(RNAi)* compared to *skn-1(RNAi)*?

– Figure 6B: draw thresholds of significance on panel B for each QTL analysis.

– The plots in C-F do not bring much except for the direction of the effect of the peaks in B. Maybe it would be more useful to provide the direction of the QTL effect in panel B either with +/- or by using a negative scale when appropriate. As we interpret it, the QTL for *mom-2(RNAi)* on chr II (if significant) is transgressive.

4) Correlation

– Figure 7: as noted by the authors, Panel B cannot be interpreted because of linkage disequilibrium. So please remove this panel and correct for LD (for example using Plink to prune SNPs under LD).

–Correlation across the genome: If we understand well, the authors use the full RAD data in Andersen et al., 2012. So a large fraction of the SNPs come from QX1211 and a few other isolates. we suspect that the negative correlation may be driven by a small number of isolates contributing most SNPs. As this is a central result, please test whether it is robust to outliers, and genetic relatedness among lines, for example by bootstrapping on the lines.

The result, if true, implies a strong negative genetic correlation, which could be measured using a multivariate animal model (e.g., see R package sommer for worked examples).

5) *rict-1, mig-5, plp-1* loci

The rationale for testing these three loci is unclear – they appear to be cherry-picked to find a different effect on *mom-2* vs *skn-1* penetrance. Are there instead known loci that would have similar effects on *mom-2* vs *skn-1* penetrance?

6) Subsection “Phylogenetic and geographical analyses”: examining the correlation between phenotypes and phylogeny using a Mantel test seems less accurate (e.g., https://www.ncbi.nlm.nih.gov/pubmed/20163450) than simply measuring whether the phenotypes have phylogenetic signal (this can be done by estimating Pagel's lamda or Blomberg's K). Please report the phylogenetic signal of the phenotypes and remove the Mantel test.

7) Subsection “Extensive natural cryptic variation in the requirement for SKN-1 in endoderm specification within the *C. elegans* species” paragraph three: The explanation for the difference in results between study by Paaby et al. and this study appear hand-wavy. The authors should modify or provide further justification of their conclusion.

8) The language / terminology, especially in the Introduction, needs to be more precise. Here are some examples:

Introduction first sentence: change "has been bequeathed to" to "is conserved in"

Introduction fourth sentence: you could delete this sentence as the question posed is rather too broad and will not be answered by a single study focused on a single GRN.

Introduction second paragraph: "in the most ancient creatures" – the animals to which you are referring are extant. Please rephrase.

Introduction paragraph five: "shows striking variation even in relatively most closely related species" – what's "striking" and what's not seems highly subjective so consider to tone down this statement (e.g., "substantial" is a lot more neutral). Also, i would prefer if you said "between species that diverged 20-40 million years ago", which is more precise than the ambiguous "closely related".

Introduction paragraph six: delete "the radiation of" – not necessary.

– replace "profoundly" with a more neutral word (e.g., "highly").

– "exceedingly rapid" – not at all clear why this is so.

– replace "during the radiation of" with "within".

9) Length of Discussion: the discussion was rather long, largely because the authors repeated their key results. Please replace the paragraphs (e.g., subsection “Multigenic variation in the requirement for SKN-1 and MOM-2” restating your key results with single sentence summaries.

10) Content of Discussion: One interesting aspect of this study that was not discussed in detail is the question whether this level of regulatory plasticity is restricted to a few, fast evolving taxa such as *C. elegans* and *Drosophila* or whether it is more widespread throughout metazoa. Please brief discuss this aspect of your results. In contrast, the data shown in this study do not provide definitive support for the hourglass hypothesis as the authors have argued for and this type of plasticity might be restricted to a few fast evolving taxa and not a defining characteristic feature of metazoan development. Please de-emphasize or altogether remove discussion of the developmental hourglass model as the data do not directly address it.

11) Figure 9 showing the simplified models should be expanded to include the additional factors *rict-1, plp-1* and *mig-5* so that it provides a more complete representation of the data.

12) Title: Maybe add "MOM-2"/Wnt since only *mom-2* has been tested. The variation in requirement for *mom-2* could in principle be due to variation in redundancy with another Wnt ligand. Also, "ancient": what is ancient? the network or only the final TF?

---

## [Author Response]

Essential revisions:1) Statements on RNAi sensitivity (e.g. third paragraph of subsection “Extensive natural cryptic variation in the requirement for SKN-1 in endoderm specificationwithin the C. elegans species”: "fully sensitive" to RNAi). Please make clear that sensitivity to RNAi is a quantitative trait. Although the authors place themselves in very strong RNAi conditions that result in 100% lethality (in contrast to a previous study by Paaby et al., 2015 where RNAi was weaker), they cannot conclude that RNAi efficiency is the same in all isolates or RILs. It may be that 0-5% of expression of the target gene causes 100% lethality, but a range of endoderm phenotypes. The introgression results are convincing for the tested isolates; however, it remains possible that some genetic variation for RNAi efficiency is detected. Of course, it would not explain the negative correlation between the two RNAi experiments, which may be worth to point out.

We agree that variations in RNAi sensitivity among wild isolates could confound our analyses and we have added statements in the revised text to clarify this point. However, as we note in the text, for *skn-1(RNAi)*, RNAi in the N2 strain produced essentially identical results as the strong loss-of-function *skn-1(zu67)* allele. This means that the 44 strains that showed a *stronger* effect than N2 (i.e., a smaller fraction of embryos with endoderm) cannot be explained by differences in RNAi alone but must reflect *bona fide* genetic variation affecting the endoderm GRN. In the revised manuscript, we have also included a statement indicating that it remains possible that genetic variation in RNAi efficiency could underlie some of the variation observed (particularly for the strains with weaker effects than N2), which could affect the phenotypes measured (noted in paragraph five of subsection “Extensive natural cryptic variation in the requirement for SKN-1 in endoderm specification within the C. elegans species”). In this regard, we note in the revised text that variation in RNAi could explain the weak positive correlation across the wild isolates between variants underlying the *skn-1(RNAi)* and *mom-2(RNAi)* phenotypes (subsection “Potential cryptic relationships between SKN-1 and MOM-2 inputs”) (Figure 7—figure supplement 1). On the other hand, as you have pointed out, the genome-wide negative correlation we observed between the two RNAi phenotypes across the RILs (Figure 7B) cannot be readily explained by variation in RNAi efficacy. Thus, variation in the GRN *per se* must underlie the key findings we report.

2) Comments on GWAS analyses:– We suggest that the authors only show the EMMA results. Is there a reason to use a model that does not correct for population structure?

We have removed all non-population corrected GWAS analyses from the manuscript, as suggested.

(NB: EMMA performs a GWAS, so do not refer to GWAS and EMMA as two different things.)

We had kept EMMA as a separate nomenclature to avoid confusion between the two analyses. However, as suggested, we have removed the non-population-corrected GWAS, and referred to all GWAS analyses correctly.

– Since the phenotypes are proportions, consider to use an arcsine transformation or best a binomial model on individual observations taking replicates into account. This may particularly help for the skewed mom-2 data.

Prior to initial submission, we transformed the *mom-2* phenotypic values by a variety of approaches in an effort to obtain a more normal distribution; however, none of these treatments led to a normal distribution. The results of those analyses were not included as they did not differ in any substantial way from the ones presented in the paper.

As for the use of a binomial model, this was also tested in the *skn-1(RNAi)* data in the non-population-adjusted GWAS analysis, which gave identical significant SNPs to those reported in the original manuscript. We argue that this reflects the high reproducibility of the phenotype: we found virtually no variability between biological replicates, even between different researchers (6 at the time of submission) in different countries (the US and New Zealand). For the *mom-2(RNAi)* data, we did not find any significant SNPs, but the results were slightly different as there was small variability between replicates for this phenotype.

As there are no substantial differences with these other analyses, for the sake of clarity, and as suggested by the reviewers, we report only the population-corrected GWAS results in the revised version, using the original data (non-transformed).

– Where does the EMMA peak on chromosome IV with skn-1(RNAi) map? Please provide the association results in a supplementary table. Since the skn-1 gene is in this region (as far as can be judged), can you rule out that variation in RNAi efficiency due to skn-1 polymorphism underlies it?

We have provided a table with the significant SNPs for *skn-1(RNAi)* as Table 1. *skn-1* is located on Ch IV: 5,651,087-5,660,523. The SNPs most strongly associated are just outside this region (and are not in LD). However, it should be noted that a causal region for MY16 is closely linked to *skn-1*, as shown in the *zu67* introgression results (Figure 2C).

There are only three polymorphisms found in the *skn-1 gene* (shown in Author response table 1). The only SNP that could potentially lead to a change is the one at position 5,655,535, which is found in wild isolates JU751, MY23, and MY16. This missense mutation is situated away from the DNA binding domain and therefore should not have a large impact on SKN-1 function. But most importantly, as has been documented in previous studies, a small number of mismatches are generally well-tolerated in RNAi [1]. Furthermore, RNAi is fully effective in the MY16 genetic background as we show in Figure 2C. Therefore, it would appear that variation in SKN-1 dependence is not likely attributable to variation in RNAi efficiency resulting from underlying *skn-1* polymorphisms.

**Author response table 1. resptable1:** 

*position*	*Ref/Alt*	*Wild isolates with alt variant (used in this study)*
*IV:5,653,761*	*C/T*	*QX1211*
*IV:5,654,947*	*C/T*	*QX1211*
*IV:5,655,535*	*T/C,A*	*JU751, MY23, MY16*

3) Regarding RILs:– Is there any reason to perform marker regression versus interval mapping? Just go for interval mapping.

We agree with the reviewers that marker regression does not add any additional information and we have therefore removed marker regression in the revised manuscript.

– Please comment whether all lines show 100% lethality upon RNAi of each gene.

We observed 100% lethality in all cases, as we now state throughout the revised manuscript.

– Figure 6A: why were so few lines tested for mom-2(RNAi) compared to skn-1(RNAi)?

Our main focus for this part of the study was to understand the relationship between MOM-2 and SKN-1 function. Thus, to evaluate the MOM-2 requirement, we first tested all RILs for SKN-1requirementand, based on those results, selected a subset that represented the full span of the *skn-1(RNAi)* phenotype. This way, we were sure to cover the entire range of phenotypes without having to resort to time-consuming analysis of all ~100 lines (at ~500 embryos per line, or 50,000 embryos). Our analysis of ~1/3rd of the lines provided a strongly representative set that is sufficient for QTL mapping, as substantiated by calculations described in reference [2] and for understanding the relationship between SKN-1 and MOM-2 requirements.

– Figure 6B: draw thresholds of significance on panel B for each QTL analysis.

As suggested, we have included significance thresholds in the revised manuscript (Figure 6B).

– The plots in C-F do not bring much except for the direction of the effect of the peaks in B. Maybe it would be more useful to provide the direction of the QTL effect in panel B either with +/- or by using a negative scale when appropriate. As we interpret it, the QTL for mom-2(RNAi) on chr II (if significant) is transgressive.

We found that including the effects to panel B, as suggested, resulted in a very busy figure that became rather difficult to interpret. We therefore chose to keep it separate. Plots C-F have been switched for a combined effect plot showing the significant QTL peaks for both *skn-1* and *mom-2*, as further discussed in the revised Results. This presentation illustrates both the size and directionality of the QTL effects (Figure 6C).

4) Correlation– Figure 7: as noted by the authors, Panel B cannot be interpreted because of linkage disequilibrium. So please remove this panel and correct for LD (for example using Plink to prune SNPs under LD).–Correlation across the genome: If we understand well, the authors use the full RAD data in Andersen et al., 2012. So a large fraction of the SNPs come from QX1211 and a few other isolates. we suspect that the negative correlation may be driven by a small number of isolates contributing most SNPs. As this is a central result, please test whether it is robust to outliers, and genetic relatedness among lines, for example by bootstrapping on the lines.The result, if true, implies a strong negative genetic correlation, which could be measured using a multivariate animal model (e.g., see R package sommer for worked examples).

In response to the reviewer’s very useful suggestion, we re-analyzed our data and have included the correlation effects using a pruned SNP set (using PLINK) and z-score (to remove outliers) in our new calculations of allelic effects. While re-evaluating the correlations, we discovered an error in the original code which, when corrected, revealed a more complex relationship across the wild isolates that is robust to outliers and genetic relatedness. We have therefore revised the text relevant to the correlations across the wild isolates and have now included additional data on the reciprocal effects (i.e., negative correlation) in the N2 x MY16 RILs. From the latter analysis, we found a strong negative correlation between the requirements for MOM-2 and SKN-1 in the RILs across five of the chromosomes, as described in the revised text, supporting the notion of a reciprocal relationship between MOM-2 and SKN-1. However, based on our analysis of chromosome IV, this effect appears to be modified by additional genetic interactions. Although chromosome IV showed a broadly *positive* correlation between these phenotypes, we found that by constructing near-isogenic lines (NILs) that separated the variants on this chromosome, the majority, but not all, of the resulting recombinant lines showed reciprocality in the MOM-2 and SKN-1 phenotypes. We propose that this reciprocal effect, which is also supported by our analysis of the effect of *rict-1, mig-5,* and *plp-1* on the *skn-1* and *mom-2* phenotypes may be masked by additional interactions between multiple variants. Owing to this observed complexity, we have toned down the conclusions regarding reciprocality of the SKN-1 and MOM-2 requirements throughout the revised manuscript, including in the Abstract, Introduction, Results, and Discussion, and hypothesize that complex genetic interactions may mask the effect in some cases.

5) rict-1, mig-5, plp-1 lociThe rationale for testing these three loci is unclear – they appear to be cherry-picked to find a different effect on mom-2 vs skn-1 penetrance. Are there instead known loci that would have similar effects on mom-2 vs skn-1 penetrance?

As now described in the revised text, we selected these genes by searching for genes in the regions of the identified QTL regions that have been implicated in endoderm development. For *plp-1* (chromosome IV QTL for the *skn-1* phenotype), there is a predicted coding difference (Val112/Asp112), near the DNA binding domain between MY16 and N2. For *mig-5*, we identified variations in the coding regions and in the 3’ UTR. For *rict-1*, we found substantial variation in the coding region as well as many variants in the introns. Although we have not yet demonstrated a causal relationship for any of these genes, RNAi of each gave results that were consistent with compensatory effects between SKN-1 and MOM-2: i.e., a reciprocal relationship between these two inputs, further supporting this hypothesis.

6) Subsection “Phylogenetic and geographical analyses”: examining the correlation between phenotypes and phylogeny using a Mantel test seems less accurate (e.g., https://www.ncbi.nlm.nih.gov/pubmed/20163450) than simply measuring whether the phenotypes have phylogenetic signal (this can be done by estimating Pagel's lamda or Blomberg's K). Please report the phylogenetic signal of the phenotypes and remove the Mantel test.

We agree that measuring phylogenetic signal of the phenotypes is more accurate in this scenario. We have removed the Mantel test and included the phylogenetic signal by estimating Pagel’s λ in the revised manuscript. This statistic, λ, for *skn-1(RNAi)* is 0.47 (p-value = 0.14) while λ for *mom-2(RNAi)* is 6.9 x 10^-05^ (p-value = 1). These results indicate no phylogenetic signal for either phenotype: *i.e.*, close relatives are not more similar than distant relatives. These methods and results have been included in the revised text and figures (Figure 3 and Figure 4—figure supplement 1).

7) Subsection “Extensive natural cryptic variation in the requirement for SKN-1 in endoderm specification within the C. elegans species” paragraph three: The explanation for the difference in results between study by Paaby et al. and this study appear hand-wavy. The authors should modify or provide further justification of their conclusion.

In response to this concern we have included more details on this issue and justification for the differences observed in the revised text (paragraph four of subsection “Cryptic variation in the quantitative requirement for MOM-2/Wnt, but not POP-1, in endoderm development”). We note that Paaby et al. [3] examined lethality with RNAi of maternal-effect genes (including *skn-1* and *mom-2*) and not endoderm specification *per se*, so the traits evaluated are different. Nevertheless, we observed highly reproducible, penetrant (100%) lethality across the wild isolates in our experiments across many replicates. These results differ starkly from those reported by Paaby et al. in which penetrant lethality was not seen, lethality varied across wild isolates, and high variability was observed between replicates. We suggest that this difference with our findings is attributable to variability in RNAi efficacy in the Paaby et al. study, in which liquid RNAi culture was used. Liquid media is fundamentally different from the conditions experienced on agar plates and different culture conditions can result in significant differences in gene expression changes [4]. We clarify these points in the revised text (subsection “RNAi”), in which we explain these technical differences. We emphasize the highly reproducible effects and fully penetrant lethality that we have observed.

8) The language / terminology, especially in the Introduction, needs to be more precise. Here are some examples:

We agree with this suggestion and have modified the terminology throughout the text as appropriate. In the specific cases below, we have noted those individually.

Introduction first sentence: change "has been bequeathed to" to "is conserved in".

This modification has been made in the revision.

Introduction fourth sentence: you could delete this sentence as the question posed is rather too broad and will not be answered by a single study focused on a single GRN.

We have removed this sentence, as suggested.

Introduction second paragraph: "in the most ancient creatures" – the animals to which you are referring are extant. Please rephrase.

We referred here to basal metazoans that contain a digestive tract. We have replaced the sentence in the revised manuscript for clarity.

Introduction paragraph five: "shows striking variation even in relatively most closely related species" – what's "striking" and what's not seems highly subjective so consider to tone down this statement (e.g., "substantial" is a lot more neutral). Also, I would prefer if you said "between species that diverged 20-40 million years ago", which is more precise than the ambiguous "closely related".

We have changed this wording throughout the revised manuscript.

Introduction paragraph six: delete "the radiation of" – not necessary.

We have deleted this phrase, as suggested.

– replace "profoundly" with a more neutral word (e.g., "highly").

We have replaced “profoundly” with “highly”.

– "exceedingly rapid" – not at all clear why this is so.

This statement has been removed.

– replace "during the radiation of" with "within".

This wording has been replaced throughout the revised manuscript, as suggested.

9) Length of Discussion: the discussion was rather long, largely because the authors repeated their key results. Please replace the paragraphs (e.g., subsection “Multigenic variation in the requirement for SKN-1 and MOM-2” restating your key results with single sentence summaries.

We have revised the discussion and shortened some sections substantially.

10) Content of Discussion: One interesting aspect of this study that was not discussed in detail is the question whether this level of regulatory plasticity is restricted to a few, fast evolving taxa such as C. elegans and Drosophila or whether it is more widespread throughout metazoa. Please brief discuss this aspect of your results. In contrast, the data shown in this study do not provide definitive support for the hourglass hypothesis as the authors have argued for and this type of plasticity might be restricted to a few fast evolving taxa and not a defining characteristic feature of metazoan development. Please de-emphasize or altogether remove discussion of the developmental hourglass model as the data do not directly address it.

We thank the reviewer for the suggestion. We have now included discussion on high divergence of early developmental events observed across different taxa, including mouse, fish, frogs, flies and worms. We note that this may be due to weak purifying selection of maternal-effect genes (paragraph three subsection “Cryptic variation and evolvability of GRNs”). We have de-emphasized the developmental hourglass model in our revised manuscript. Although our results do not provide definitive support for the hourglass hypothesis, rapid changes in early embryonic regulations are one of the main features of the hourglass model, and we felt obligated to discuss that at least briefly in the revised manuscript.

11) Figure 9 showing the simplified models should be expanded to include the additional factors rict-1, plp-1 and mig-5 so that it provides a more complete representation of the data.

The figure has been modified as suggested by the reviewers.

12) Title: Maybe add "MOM-2"/Wnt since only mom-2 has been tested. The variation in requirement for mom-2 could in principle be due to variation in redundancy with another Wnt ligand. Also, "ancient": what is ancient? the network or only the final TF?

We have changed the title to “Extensive intraspecies cryptic variation in an ancient embryonic gene regulatory network.” We chose to keep the title more general to better reflect the most important findings of this work, which is the extensive variation found in the inputs into the endoderm GRN. We have also revised the text to make it clear that we are referring to the ancient nature of the endoderm and the conserved GRN components that control it as has been articulated in published literature that we have cited.

References:

1. Saxena S, Jónsson ZO, Dutta A. Small RNAs with imperfect match to endogenous mRNA repress translation. Implications for off-target activity of small inhibitory RNA in mammalian cells. J Biol Chem. 2003;278: 44312–44319.

2. Vales MI, Schön CC, Capettini F, Chen XM, Corey AE, Mather DE, et al. Effect of population size on the estimation of QTL: a test using resistance to barley stripe rust. Theor Appl Genet. 2005;111: 1260–1270.

3. Paaby AB, White AG, Riccardi DD, Gunsalus KC, Piano F, Rockman MV. Wild worm embryogenesis harbors ubiquitous polygenic modifier variation. Elife. 2015;4. doi:10.7554/eLife.09178

4. Çelen İ, Doh JH, Sabanayagam CR. Effects of liquid cultivation on gene expression and phenotype of C. elegans. BMC Genomics. 2018;19: 562.